# Efficient Submodular Maximization for Sums of Concave over Modular Functions

**Yang Lv**, **Guihao Wang**, **Dachuan Xu**, and **Ruiqi Yang** [*]

Institute of Operations Research and Information Engineering
Beijing University of Technology
Beijing 100124, P.R. China
{lvyang,Wangghao}@emails.bjut.edu.cn, {xudc,yangruiqi}@bjut.edu.cn

## Abstract

Submodular maximization has broad applications in machine learning, network design, and data mining. However, classical algorithms often suffer from prohibitively high computational costs, which severely limit their scalability in practice. In this work, we focus on maximizing Sums of Concave over Modular functions (SCMs), an important subclass of submodular functions, under three fundamental constraints: cardinality, knapsack, and partition matroids. Our method integrates three components: continuous relaxation, Accelerated Approximate Projected Gradient Ascent (AAPGA), and randomized rounding, to efficiently compute near-optimal solutions. We establish a $(1 - \varepsilon - \eta - e^{-\Omega(\eta^2)})$ approximation guarantee for both cardinality and partition matroid constraints, with query complexity $O\big(n^{1/2}\varepsilon^{-1/2}(T_1 + T_2)\big)$. For the knapsack constraint, the approximation ratio degrades by a factor of $1/2$, with query complexity $O\big(nT_1 + n^{1/2}\varepsilon^{-1/2}T_2\big)$, where $T_1$ denotes the computational cost of evaluating the concave extension, and $T_2$ denotes the computational cost of backpropagation. By leveraging efficient convex optimization techniques, our approach substantially accelerates convergence toward high-quality solutions. In empirical evaluations, we demonstrate that AAPGA consistently outperforms standard PGA. On small-scale experiments, AAPGA achieves superior results in significantly less time, being up to $32.3\times$ faster than traditional methods. On large-scale experiments, our parallel multi-GPU implementation further enhances performance, demonstrating the scalability of our approach.

## 1 Introduction

Submodular functions have attracted considerable attention due to their diminishing marginal returns property and have been applied in various fields such as determinantal point processes (Gillenwater et al., 2012), sensor placement (Krause et al., 2008), and probabilistic inference (Djolonga & Krause, 2014). As set functions, submodular functions are defined over the power set, with their range residing within a $2^n$-dimensional cone in $\mathbb{R}^{2^n}$, exhibiting rich structural properties. While the minimization of submodular functions can be solved in polynomial time (Fujishige, 2005), the maximization problem with cardinality constraints is NP-hard. However, researchers proposed a simple greedy algorithm, which guarantees a $(1 - 1/e)$ approximation ratio for monotone submodular functions under constraints (Nemhauser et al., 1978), and this ratio has been proven to be optimal (Feige, 1998). Moreover, different variants of the submodular maximization problem have been extensively studied, further advancing submodular optimization theory

Even for approximate solutions to submodular maximization, the extremely large scale of modern machine learning problems makes acceleration of submodular optimization a crucial research direction. Existing acceleration methods for submodular maximization can be broadly categorized as follows. The earliest serial acceleration approaches (Minoux, 1978; Mirzasoleiman et al., 2015)

---

[*]Corresponding author: Ruiqi Yang

Table 1: Comparison of representative algorithms for submodular maximization under different constraints. $T_1$ denotes the computational cost of evaluating the concave extension, and $T_2$ denotes the computational cost of backpropagation.

| Method | Type | Constraint | Approximation Guarantee | Query Complexity |
|---|---|---|---|---|
| *Cardinality Constraint* | | | | |
| Lazier-Greedy (Mirzasoleiman et al., 2015) | Submodular | Cardinality | $1 - 1/e - \varepsilon$ | $O\left(n \log \frac{1}{\varepsilon}\right)$ |
| FAST (Breuer et al., 2020) | Submodular | Cardinality | $1 - 1/e - \varepsilon$ | $O(n \log \log k)$ |
| LS+PGB (Chen et al., 2021) | Submodular | Cardinality | $1 - 1/e - \varepsilon$ | $O\left(\frac{n}{\varepsilon}\right)$ |
| *Matroid Constraint* | | | | |
| Subquad (Kobayashi & Terao, 2024) | Submodular | Matroid | $1 - 1/e - \varepsilon$ | $O\left(n^{3/2} \log^{3/2} n\right)$ |
| SPGA (Karimi et al., 2017) | Coverage Functions | Matroid | $1 - 1/e - \varepsilon$ | $O\left(\frac{n^2}{\varepsilon^2} T_2\right)$ |
| PGA (Bai et al., 2018) | Deep Submodular | Matroid | $1 - \varepsilon - \eta - e^{-\Omega(\eta^2)}$ | $O\left(\frac{n^2}{\varepsilon^2} T_2\right)$ |
| *Knapsack Constraint* | | | | |
| Modified-Greedy (Khuller et al., 1999) | Coverage Functions | Knapsack | $\frac{1}{2}(1 - 1/e)$ | $O(n^2)$ |
| Enum-Greedy (Sviridenko, 2004) | Submodular | Knapsack | $1 - 1/e$ | $O(n^5)$ |
| Density Greedy (Ene & Nguyen, 2019) | Submodular | Knapsack | $1 - 1/e - \varepsilon$ | $O\left((1/\varepsilon)^{O(1/\varepsilon^4)} n \log^2 n\right)$ |
| ParSKP (Cui et al., 2023) | Submodular | Knapsack | $\frac{1}{\sqrt{5}+2} - \varepsilon$ | $O(nk \log^2 n)$ |
| MultiStream (Cui et al., 2025) | Submodular | Knapsack | $\frac{1}{2} - \varepsilon$ | $O\left(\frac{n}{\varepsilon} \log \frac{1}{\varepsilon}\right)$ |
| *Our Proposed Method* | | | | |
| **Ours** | SCM | Cardinality / Partition Matroid | $1 - \varepsilon - \eta - e^{-\Omega(\eta^2)}$ | $O\left(n^{1/2} \varepsilon^{-1/2}(T_1 + T_2)\right)$ |
| **Ours** | SCM | Knapsack | $\frac{1}{2}\left(1 - \varepsilon - \eta - e^{-\Omega(\eta^2)}\right)$ | $O\left(nT_1 + n^{1/2} \varepsilon^{-1/2} T_2\right)$ |

avoid evaluating the marginal gains of all elements at each step, while still achieving utility comparable to the standard greedy algorithm. However, their $O(n)$ serial complexity remains inadequate for large-scale applications. Subsequently, researchers have developed streaming algorithms (Badanidiyuru et al., 2014; El Halabi et al., 2020), parallel algorithms (Breuer et al., 2020; Chen et al., 2021; Cui et al., 2023; Amanatidis et al., 2021), distributed algorithms (Mirzasoleiman et al., 2016), continuous relaxations (Badanidiyuru & Vondrák, 2014), and multi-stage submodular maximization methods (Wei et al., 2014), among others.

Many of these acceleration algorithms treat the submodular function as a black box, adopting the *value oracle model*. While such methods can improve computational efficiency, they do not fully exploit the structural properties of submodular functions. In recent years, more targeted studies on specific function classes have achieved order-of-magnitude speedups. For example, (Iyer & Bilmes, 2019) introduced memoization framework to maintain auxiliary information of sets, enabling rapid evaluation of submodular functions and representing one of the first works to leverage internal structure for optimization. (Karimi et al., 2017) studied coverage functions and proposed a "three-step" framework: first relaxing the function continuously, then solving the constrained continuous optimization problem, and finally applying rounding to obtain an integer solution. By exploiting the specific properties of coverage functions, this approach significantly reduces computational cost. The introduction of deep submodular functions (Dolhansky & Bilmes, 2016) enabled neural networks to capture a broader class of submodular objectives, further facilitating efficient GPU-based optimization (Bai et al., 2018). Nevertheless, the number of iterations required by Projected Gradient Ascent remains $O\left(n^2/\varepsilon^2\right)$, with query complexity $O\left(\left(n^2/\varepsilon^2\right) T_2\right)$, where $T_2$ denotes the computational cost of backpropagation. This remains prohibitively large for large-scale problems. More detailed results and contributions are provided in Table 1. The combination of submodularity and neural networks could open new avenues for integrating learning-based techniques with classical optimization theory, and its effectiveness has already begun to be demonstrated in applications (Ghadimi & Beigy, 2020; Jaisankar et al., 2024).

In this work, we study the maximization of Sums of Concave over Modular functions (SCMs) and propose an efficient optimization framework inspired by neural networks. We design an approximation algorithm with query complexity $O\left(n^{1/2} \varepsilon^{-1/2}(T_1 + T_2)\right)$ for cardinality and partition matroid constraints, and $O\left(nT_1 + n^{1/2} \varepsilon^{-1/2} T_2\right)$ for the knapsack constraint, where $T_1$ denotes the computational cost of evaluating the concave extension and $T_2$ denotes the cost of backpropagation. In the traditional value oracle model, costs such as $T_1, T_2$ are often neglected, as the oracle is typically treated as a "black box," where the computational process is considered a "zero-cost" operation. The framework integrates three key components: continuous relaxation, Accelerated Approximate Projected Gradient Ascent (AAPGA), and randomized rounding. Our method achieves an approximation ratio of $\max_{\eta \in [0,1]}(1 - \varepsilon)(1 - \eta)\left[1 - |V| \exp\left(-\eta^2 \min_{i \in V} \frac{\min m_i \cdot k}{\max m_i}\right)\right]$ under cardinality and partition

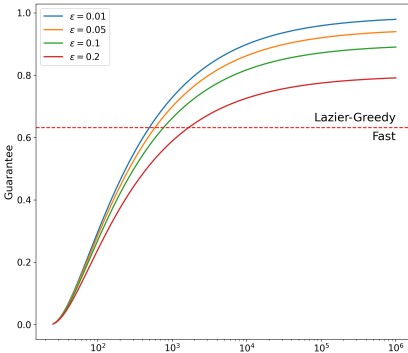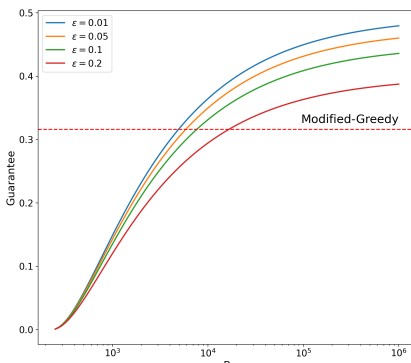

Figure 1: The figure shows the trends of the approximation ratios for cardinality and partition matroid constraints as $k$ varies, as presented in Theorem 2. The left plot assumes $\min m_i / \max m_i = 0.1$ and $|V| = 10$. The right plot shows the trend of the approximation ratio for the knapsack constraint as the knapsack capacity $B$ changes, with $\min m_i / \max m_i = 0.1$, $\max c_i = 10$, and $|V| = 10$.

matroid constraints, and $\frac{1}{2} \max_{\eta \in [0,1]} (1-\varepsilon)(1-\eta) \left[ 1 - |V| \exp \left( -\eta^2 \min_{i \in V} \frac{\min m_i \cdot B}{\max m_i \max c_i} \right) \right]$ under the knapsack constraint. The comparison of approximation ratios with other algorithms is shown in Figure 1. As can be seen, with the increase of the constraints $k$ and $B$, the approximation ratio tightens accordingly, outperforming the algorithms presented in the figure. This suggests that our method provides better theoretical guarantees for large-scale submodular optimization. Compared with the algorithm of Bai et al. (2018), which requires $O(n^2/\varepsilon^2 T_2)$ time, our approach offers substantially stronger computational guarantees.

In the experimental evaluation, we compare our algorithm with existing submodular/SCM maximization algorithms. Under cardinality and knapsack constraints, AAPGA achieves a speedup of up to 32.3 times while maintaining strong solution quality. On large-scale datasets, our multi-GPU parallelization implementation further outperforms PGA with pipage rounding, demonstrating both faster convergence and practical scalability.

## 2 BACKGROUND AND PROBLEM SETUP

The submodular function $f$ is a set function defined on $\mathcal{N} = \{1, 2..., n\}$. It satisfies the diminishing returns property, i.e., for any $A \subseteq B \subseteq \mathcal{N}$ and any $e \in \mathcal{N} \setminus B$, it holds that $f(A \cup \{e\}) - f(A) \geq f(B \cup \{e\}) - f(B)$. In this paper, we only consider monotone submodular functions, i.e., for $A \subseteq B \subseteq \mathcal{N}$, we have $f(A) \leq f(B)$.

**Definition 1 (SCM).** A *Sum of Concave over Modular functions* (SCM) is defined as follows. Let $\{\phi_i : \mathbb{R}_{\geq 0} \to \mathbb{R}_{\geq 0}\}_{i \in V}$ be a collection of non-negative, non-decreasing monotone, and normalized concave functions with $\phi_i(0) = 0$. Let $\{m_i : 2^{\mathcal{N}} \to \mathbb{R}_{\geq 0}\}_{i \in V}$ be the corresponding modular functions defined by $m_i(A) = \sum_{v \in A} m_i(v), A \subseteq \mathcal{N}$. Then, the function

$$f(A) = \sum_{i \in V} \phi_i(m_i(A)), \quad A \subseteq \mathcal{N} \tag{1}$$

is called an SCM. It is monotone and submodular.

The modular functions $m_i$ in equation 1 can be naturally extended to linear mappings over the continuous domain, i.e., $m_i : [0,1]^n \to \mathbb{R}_+$. Since the composition of a concave function with a linear function remains concave, and the sums of concave functions is also concave, we can construct a concave extension $F(x)$ of $f(A)$ such that $f(A) = F(\mathbf{1}_A)$ for every $A \subseteq \mathcal{N}$. Formally, this extension is given by

$$F(\mathbf{x}) = \sum_{i \in V} \phi_i\big(m_i(\mathbf{x})\big), \quad \mathbf{x} \in [0,1]^n. \tag{2}$$

Previous studies on submodular maximization via continuous relaxation typically rely on the multilinear extension (Calinescu et al., 2011), defined as $F_m(\mathbf{x}) = \sum_{A \subseteq \mathcal{N}} f(A) \prod_{v \in A} x_v \prod_{v \in \mathcal{N} \setminus A} (1 -$

$x_v$), which can be equivalently written as $F_m(\mathbf{x}) = \mathbb{E}_{A \sim \mathcal{D}_\mathbf{x}}[f(A)]$, $\Pr(A) = \prod_{v \in A} x_v \prod_{v \in V \setminus A}(1 - x_v)$. The multilinear extension enjoys a directional concavity property, which makes it particularly suitable for rounding schemes that yield strong approximation guarantees.

However, $F_m(x)$ is generally intractable to compute exactly, and sampling-based approximations are often required. In practice, evaluating its gradient involves $\frac{\partial F_m(\mathbf{x})}{\partial x_i} = F_m(\mathbf{x} \vee \mathbf{e}_i) - F_m(\mathbf{x} \wedge \bar{\mathbf{e}}_i)$, which must also be approximated via stochastic sampling. In contrast, the concave extension does not possess the same directional concavity that facilitates rounding, but it has two important advantages: (i) it can be efficiently parallelized on GPUs (including multi-GPU settings), and (ii) supergradients can be efficiently obtained via backpropagation. Therefore, in our approach we adopt the concave extension within the AAPGA optimization stage to accelerate computation, while leveraging the multilinear extension in the rounding step to provide theoretical guarantees. This hybrid strategy combines the computational efficiency of the concave extension with the approximation quality enabled by the multilinear extension, leading to both faster optimization and provable performance bounds.

SCMs can express a wide variety of functions. For the classic coverage function (Feige, 1998), let the element set be $U = \{e_1, e_2, ..., e_n\}$, and let there be $m$ subsets $s_1, s_2, ..., s_m \subseteq U$. For any index set $A \subseteq \{1, 2, ..., m\}$, we define $f(A) = \left| \bigcup_{i \in A} s_i \right|$, which can be written as an SCM in the following form: $f(\mathbf{x}) = \sum_{e_i \in E} \min\{m_{e_i}(\mathbf{x}), 1\}$, where the linear function $m_{e_i}$ assigns coefficient 1 to the $j$-th component if $s_j$ covers $e_i$, and 0 otherwise. Since $\min\{1, x\}$ is a concave function, this fits into the SCM framework. For the probabilistic coverage function (Liu et al., 2023; Dai et al., 2025), assume $s_1, s_2, ..., s_m \subseteq U$, but the probability that $s_i$ covers $e_j$ becomes $p_{ij}$. Each element $e_j$ has an associated reward $v_j$. Given an index set $A$, the expected reward of an element being covered by at least one set is: $f(\mathbf{x}) = \sum_{j=1}^{N} v_j \left(1 - \prod_{i \in [m]}(1 - p_{ij})^{x_i}\right)$. Taking the logarithm gives an equivalent form: $f(\mathbf{x}) = \sum_{j=1}^{N} v_j \left(1 - \exp\left(\sum_{i=1}^{M} x_i \log(1 - p_{ij})\right)\right)$, where the function $g(x) = 1 - \exp(-cx)$, for $c > 0$, is concave. Hence, probabilistic coverage is an instance of SCM. The facility location function (Wei et al., 2015) is also an SCM. It selects a subset $A$ from the demand set $U$ such that the facilities in $A$ provide maximum benefit. The overall service quality is given by: $f(A) = \sum_{u \in U} \max_{a \in A} w(a, u)$, which can be approximated using the following expression: $\phi_\gamma(\mathbf{x}) = \sum_{u \in U} \frac{1}{\gamma} \log\left(\sum_{i=1}^{n} x_i \exp(\gamma w_a)\right)$. As $\gamma$ approaches infinity, $\phi_\gamma(\mathbf{x})$ matches $f(A)$ at integer points, and $\phi_\gamma(\mathbf{x})$ is an SCM function.

In submodular maximization, typical constraints include the cardinality constraint $|A| \leq k$, the knapsack constraint $c(A) \leq B$, where $c(A) = \sum_{i \in A} c_i$ denotes the total cost of selecting set $A$ under item costs $c_i$, and the partition matroid constraint $|A \cap G_j| \leq r_j$, $\forall j \in [m]$, where the ground set $\mathcal{N}$ is partitioned into disjoint groups $G_1, G_2, \ldots, G_m$, each with its own rank bound $r_j$. These can be unified under a general optimization framework:

$$\max_{A \in \mathcal{C}} f(A), \tag{3}$$

where $\mathcal{C}$ denotes the feasible region specified by the chosen constraint.

## 3 OPTIMIZATION FOR SCM MAXIMIZATION

We follow the optimization framework proposed by (Bai et al., 2018). In this framework, the problem is first extended to the continuous domain via SCM, followed by the computation of supergradients. Then, the Projected Gradient Ascent algorithm is applied to obtain a continuous solution, which is subsequently rounded to an integral solution using pipage rounding. Based on this framework, we make the following improvements: (i) we extend the computation of supergradients to the case of non-differentiable SCMs; (ii) we replace Projected Gradient Ascent with the Accelerated Approximate Projected Gradient Ascent method; and (iii) we provide a more detailed discussion of rounding techniques under cardinality, knapsack and partition matroid constraints, along with proofs of their approximation guarantees and running time complexities.

**Continuous Relaxation** We observe that several commonly used functions, such as $\min\{x, 1\}$, $1 - \exp(x)$, $\log(1 + x)$, and $\sqrt{x}$, are concave and can serve as activation functions. However, these functions may not be differentiable at certain points. Therefore, it is necessary to explain how to

perform differentiation of SCMs under non-differentiable activation functions. For analysis, we consider the right derivative of $\phi$. We define the superdifferential of $F$ at $\mathbf{x}$ by $\partial F(\mathbf{x}) = \{g \in \mathbb{R}^n \mid F(\mathbf{y}) - F(\mathbf{x}) \leq g \cdot (\mathbf{y} - \mathbf{x}), \ \forall \mathbf{y} \in \mathcal{P}\}$. Any $g \in \partial F(\mathbf{x})$ is called a supergradient of $F$ at $\mathbf{x}$. The vector $g = \left(\sum_{i \in |V|} m_{ij} \, \phi'_{i+}(m_i(\mathbf{x}))\right)_j$ is an element of $\partial F(\mathbf{x})$, where $\phi'_{i+}$ denotes the right-hand derivative of $\phi$ with respect to the $i$-th share. This formulation enables efficient computation of supergradients via the backpropagation algorithm of deep neural networks, leveraging multi-GPU parallelization. It effectively addresses the inefficiency of gradient computation in the multilinear extension, which is a bottleneck in the standard continuous greedy algorithm.

Constraints originally defined over the discrete domain $\{0,1\}^n$ can be naturally relaxed to the continuous domain $[0,1]^n$. For example, the cardinality constraint of selecting at most $k$ elements from $\mathcal{N} = \{1, \ldots, n\}$, $\{\mathbf{x} \in \{0,1\}^n \mid \sum_{i=1}^n x_i \leq k\}$, is relaxed to $\{\mathbf{x} \in [0,1]^n \mid \sum_{i=1}^n x_i \leq k\}$. Similarly, the knapsack constraint with item weights $c_i$ and budget $B$, $\{\mathbf{x} \in \{0,1\}^n \mid \sum_{i=1}^n c_i x_i \leq B\}$, is relaxed to $\{\mathbf{x} \in [0,1]^n \mid \sum_{i=1}^n c_i x_i \leq B\}$. Finally, letting $\mathcal{N} = \{1, \ldots, n\}$ be partitioned into disjoint blocks $G_1, \ldots, G_m$ with capacities $r_j \in \mathbb{Z}_{\geq 0}$, the partition matroid constraint $\{\mathbf{x} \in \{0,1\}^n \mid \sum_{i \in G_j} x_i \leq r_j, \ \forall j \in [m]\}$ is relaxed to $\{\mathbf{x} \in [0,1]^n \mid \sum_{i \in G_j} x_i \leq r_j, \ \forall j \in [m]\}$. For simplicity of notation, we refer to the relaxed constraint polytope as $\mathcal{P}$ throughout the paper.

**Accelerated Approximate Projected Gradient Ascent** The AAPGA method is similar to the standard PGA method, with the key distinction that the projection is performed on $\mathbf{y}^{(t)}$, which is a linear combination of $\mathbf{x}^{(t)}$ and $\mathbf{x}^{(t-1)}$, rather than directly on $\mathbf{x}^{(t)}$. Since the projection is computed only approximately, we define the approximate projection set as

$$P'_L(\mathbf{y}^{(t)}) = \left\{ \mathbf{x}' \in \mathcal{P} \ \middle| \ \|\mathbf{x}' - \Pi_{\mathcal{P}}(\mathbf{y}^{(t)} + \tfrac{1}{L}\, \gamma_F(\mathbf{y}^{(t)}))\| \leq \delta \right\},$$

where $\Pi_{\mathcal{P}}(\cdot)$ denotes the exact projection onto the polytope $\mathcal{P}$, $\delta > 0$ is the admissible approximation error, and $\gamma_F(\mathbf{y}^{(t)}) \in \partial F$ denotes a subgradient of $F$ at $\mathbf{y}^{(t)}$.

We assume that $p'_L(\mathbf{x}) \in P'_L(\mathbf{x})$, i.e., it is one representative point from the approximate projection set. Under this formulation, the per-iteration computational cost remains nearly identical to that of the standard PGA method, while the total number of iterations required is significantly reduced. The pseudocode is given in Algorithm 1. The quadratic surrogate function is defined as $Q_L(\mathbf{x}, \mathbf{y}) = F(\mathbf{y}) + \langle \mathbf{x} - \mathbf{y}, \gamma_F(\mathbf{y}) \rangle - \frac{L}{2}\|\mathbf{x} - \mathbf{y}\|^2$.

Similar to the classical FISTA algorithm (Beck & Teboulle, 2009), our AAPGA method also achieves an $O(n/T^2)$ convergence rate. The key difference is that, due to the $\delta$-approximate projection in AAPGA, we require $\delta$ to be sufficiently small (specifically $\delta < O(1/T^3)$) to ensure convergence.

**Theorem 1.** *Suppose $F$ is continuously differentiable with Lipschitz continuous gradient, with Lipschitz constant denoted by $L(f)$, i.e., $\|\nabla f(\mathbf{x}) - \nabla f(\mathbf{y})\| \leq L(f)\|\mathbf{x} - \mathbf{y}\|, \quad \forall \mathbf{x}, \mathbf{y} \in \mathbb{R}^n$. Let $\mathbf{x}^* = \arg\max_{\mathbf{x} \in \mathcal{P}} F(\mathbf{x})$ be the optimal solution, and let $\mathbf{x}^{(T)}$ denote the output of AAPGA after $T$ iterations. If the error parameter in AAPGA satisfies $\delta < O(1/T^3)$, then the algorithm achieves*

$$F(\mathbf{x}^*) - F(\mathbf{x}^{(T)}) \ \leq \ O\!\left(\tfrac{n}{T^2}\right). \tag{8}$$

We show in the appendix that for cardinality, knapsack, and partition matroid constraints, an approximate projection $x'$ can be computed in $O(n \log(n/\delta))$ time such that $\|x' - \Pi(x)\| < \delta$.

By leveraging the faster convergence of AAPGA over the standard PGA, we obtain an improved $(1 - \varepsilon)$-approximation guarantee. Given the time complexity $T = O(n^2/\varepsilon^2)$ in (Bai et al., 2018), our method achieves a significantly faster convergence rate for the SCM maximization problem.

**Corollary 1.** *Assume that $F$ is continuously differentiable with a Lipschitz continuous gradient. For any $0 < \varepsilon < 1$, if AAPGA selects $\delta < O(\varepsilon^{3/2} n^{-3/2})$, then*

$$F(\mathbf{x}^{(T)}) \ \geq \ (1 - \varepsilon)\, F(\mathbf{x}^*).$$

*The number of iterations required by AAPGA is $O\!\left(n^{1/2}\varepsilon^{-1/2}\right)$, and the query complexity is $T \cdot (T_1 + T_2) = O\!\left(n^{1/2}\varepsilon^{-1/2}(T_1 + T_2)\right)$, where $T_1$ denotes the computational cost of evaluating the concave extension, and $T_2$ denotes the computational cost of backpropagation.*

---

**Algorithm 1** Accelerated Approximate Projected Gradient Ascent

---

**Require:** SCM concave extension $F$, constraint polytope $\mathcal{P}$, maximum iterations $T$, initial parameter $\alpha_1 = 1$, step size $L_0 > 0$, scaling factor $\beta > 1$.
1: Initialize $\mathbf{x}^{(0)} \leftarrow \arg\min_{\mathbf{x} \in \mathcal{P}} \|\mathbf{x}\|_2^2$.
2: Set $\mathbf{y}^{(1)} \leftarrow \mathbf{x}^{(0)}$.
3: **for** $t = 1, 2, \ldots, T$ **do**
4:     The superdifferential $\partial F(\mathbf{y}^{(t)})$ is obtained via backpropagation.
5:     Find the smallest non-negative integer $i_t$ such that $\bar{L} = \beta^{i_t} L_{t-1}$ satisfies

$$F\left(p'_{\bar{L}}(\mathbf{y}^{(t)})\right) \geq Q_{\bar{L}}\left(p'_{\bar{L}}(\mathbf{y}^{(t)}), \mathbf{y}^{(t)}\right). \tag{4}$$

6:     Set $L_t = \eta^{i_t} L_{t-1}$ and update

$$\mathbf{x}^{(t)} \leftarrow p'_{L_t}\left(\mathbf{y}^{(t)}\right). \tag{5}$$

7:     Update the momentum parameter

$$\alpha_{t+1} = \frac{1 + \sqrt{1 + 4\alpha_t^2}}{2}. \tag{6}$$

8:     Extrapolate

$$\mathbf{y}^{(t+1)} = \mathbf{x}^{(t)} + \frac{\alpha_t - 1}{\alpha_{t+1}}\left(\mathbf{x}^{(t)} - \mathbf{x}^{(t-1)}\right). \tag{7}$$

9: **end for**
10: **return** $\mathbf{x}^{(T)}$.

---

*Proof.* From Theorem 1, we have $F(\mathbf{x}^*) - F(\mathbf{x}^{(T)}) \leq O(n/T^2)$. To guarantee that $F(\mathbf{x}^{(T)}) \geq (1 - \varepsilon) F(\mathbf{x}^*)$, it suffices to require $O(n/T^2) \leq \varepsilon F(\mathbf{x}^*)$. This holds when, $T = O(n^{1/2}\varepsilon^{-1/2})$. Since Theorem 1 further requires $\delta < O(1/T^3)$, the projection accuracy parameter $\delta$ must satisfy $\delta \leq O(n^{-3/2}\varepsilon^{3/2})$.

The algorithm in Algorithm 1 computes the supergradient in Step 4 $T$ times, with each backpropagation step for the supergradient having a time complexity of $T_2$, resulting in a query complexity of $O(T \cdot T_2)$. In Step 5 of the algorithm in Algorithm 1, each iteration requires verifying whether $i_t = 0$ holds. The number of evaluations of $F\left(p'_{\bar{L}}(\mathbf{y}^{(t)})\right)$ and $Q_{\bar{L}}\left(p'_{\bar{L}}(\mathbf{y}^{(t)}), \mathbf{y}^{(t)}\right)$ is constant, requiring a query complexity of $O(T \cdot T_1)$. Since $\bar{L} \geq L(f)$ holds, we can deduce that equation 4 is satisfied, and thus $\bar{L}$ will not exceed $\beta L(f)$. As $\bar{L}$ grows exponentially, the number of iterations that do not satisfy equation 4 is given by $\log\left(\frac{\beta L(f)}{L_0}\right) / \log \beta$, which is of constant order. Therefore, the overall complexity is $T \cdot T_1 + T \cdot T_2 = O\left(n^{1/2}\varepsilon^{-1/2}(T_1 + T_2)\right)$. □

**Randomized Rounding.** After obtaining a fractional solution $x$, it is necessary to round it to an integral solution. For cardinality and partition matroid constraints, one can apply the pipage rounding algorithm (Calinescu et al., 2011), which runs in $O(n)$ time. This method leverages the fact that the multilinear extension satisfies convexity along directions of the form $F_m(\mathbf{y} + t(\mathbf{e}_i - \mathbf{e}_j))$, where $\mathbf{e}_i = (0, \ldots, 0, 1, 0, \ldots, 0)$ is the standard basis vector. Consequently, the rounding preserves the approximation guarantee in expectation. However, under a knapsack constraint, achieving a $(1 - \varepsilon)$-approximation via rounding requires time complexity $O\left(n^{\lceil \varepsilon^{-4}\rceil + 1}\right)$ (Kulik et al., 2013).

To address this limitation, we propose an $O(n)$-time rounding algorithm tailored for the knapsack constraint, which achieves a constant-factor approximation of $1/2$.

Since the algorithm relies on the multilinear extension, (Bai et al., 2018) established the following relationship between the concave extension and the multilinear extension:

**Lemma 1** (cf. (Bai et al., 2018)). *Let $F$ be the concave extension induced by an SCM, and let $F_m$ denote the multilinear extension of the original submodular function. Then, for all $\mathbf{x} \in [0, 1]^n$, the following inequality holds:*

$$\max_{\eta \in [0,1]} (1 - \eta)\left[1 - |V|\exp\left(-\eta^2 \triangle(\mathbf{x})\right)\right] F(\mathbf{x}) \leq F_m(\mathbf{x}) \leq F(\mathbf{x}),$$

---

**Algorithm 2** Knapsack-Constrained Maximization of SCM Functions via Rounding

---

1: **Input:** Submodular function $f$, ground set $\mathcal{N}$, knapsack polytope $\mathcal{P}$, item weights $\mathbf{c}$
2: **Step 1: Best Singleton Selection**
3:     Identify the best singleton element: $e = \arg\max_{e \in \mathcal{N}} f(e)$
4: **Step 2: Fractional Solution via Gradient Ascent**
5:     Compute a fractional solution $\mathbf{x} \in [0, 1]^n$ using Algorithm 1
6: **Step 3: Randomized Rounding**
7: **while** more than one entry of $\mathbf{x}$ is fractional **do**
8:     Randomly select a pair $(i, j)$ such that $0 < x_i, x_j < 1$
9:     Define the direction $\mathbf{v} = \frac{1}{c_i}\mathbf{e}_i - \frac{1}{c_j}\mathbf{e}_j$
10:     Compute feasible step sizes $\theta_{\min}, \theta_{\max}$ such that both satisfy $\mathbf{x} + \theta\mathbf{v} \in \mathcal{P}$
11:     Update $\mathbf{x}$ as follows:
12:         With probability $\frac{-\theta_{\min}}{-\theta_{\min} + \theta_{\max}}$, set $\mathbf{x} \leftarrow \mathbf{x} + \theta_{\max}\mathbf{v}$
13:         Otherwise, set $\mathbf{x} \leftarrow \mathbf{x} + \theta_{\min}\mathbf{v}$
14: **end while**
15: Final adjustment: set any remaining fractional entries of $\mathbf{x}$ to 0
16: **return** $\arg\max_{\mathbf{x},e}\{f(\mathbf{x}), f(e)\}$

---

*where $\triangle(\mathbf{x}) = \min_{i \in V} \frac{m_i(\mathbf{x})}{2\max m_i}$.*

By Lemma 1, the solution $\mathbf{x}^{(T)}$ obtained by AAPGA can be directly rounded via randomized rounding. Moreover, through the multilinear extension, this procedure enjoys theoretical approximation guarantees.

**Theorem 2.** *For the SCM maximization problem under cardinality and partition matroid constraints, combining AAPGA with pipage rounding yields the following guarantee:*

$$\mathbb{E}[f(x')] \geq \max_{\eta \in [0,1]}(1 - \varepsilon)(1 - \eta)\left[1 - |V|\exp\left(-\eta^2\min_{i \in V}\frac{\min m_i \cdot k}{\max m_i}\right)\right]f(x_{\text{OPT}}). \quad (9)$$

*Under the knapsack constraint, applying Algorithm 2 with AAPGA and randomized rounding gives*

$$\mathbb{E}[f(x')] \geq \tfrac{1}{2}\max_{\eta \in [0,1]}(1 - \varepsilon)(1 - \eta)\left[1 - |V|\exp\left(-\eta^2\min_{i \in V}\frac{\min m_i \cdot B}{\max m_i \max c_i}\right)\right]f(x_{\text{OPT}}).$$

*The overall query complexity is $O\left(n^{1/2}\varepsilon^{-1/2}(T_1 + T_2)\right)$ for cardinality and partition matroid constraints, and $O\left(n \cdot T_1 + n^{1/2}\varepsilon^{-1/2}T_2\right)$ for the knapsack constraint.*

The approximation bounds above are not fixed constants; however, (Karimi et al., 2017) proved that for coverage functions, the approximation ratio reaches $1 - 1/e - \varepsilon$. This suggests that in many practical scenarios, our algorithm can achieve significantly better empirical performance than the worst-case guarantee.

## 4 EMPIRICAL EVALUATION

The numerical experiments are organized into three parts. First, we demonstrate that AAPGA achieves faster convergence than PGA under the concave extension. Second, we show that AAPGA combined with randomized rounding outperforms accelerated submodular maximization algorithms in terms of both solution quality and computational efficiency. Finally, we evaluate large-scale graph problems, where AAPGA with pipage rounding, enhanced by multi-GPU parallelization, achieves superior performance compared to PGA combined with pipage rounding. All experiments are conducted on a server equipped with a Hygon C86 7380 32-core CPU, two NVIDIA A800 80GB GPUs, and 256 GB RAM. The system was running CUDA 12.2. Our implementation is open-sourced at `https://github.com/lvymath1/Efficient-Submodular-Maximization`.

### 4.1 CONVERGENCE SPEED OF AAPGA

The convergence rate of AAPGA to achieve a $(1 - \varepsilon)$ approximation ratio is $T = O(n^{1/2}\varepsilon^{-1/2})$, while PGA requires $T = O(n^2\varepsilon^{-2})$. Due to Nesterov acceleration, AAPGA exhibits a substantial advantage over PGA.

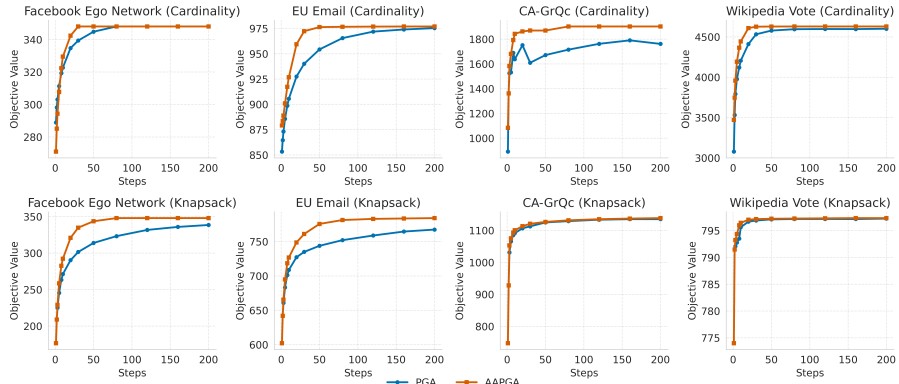

Figure 2: Performance comparison between PGA and AAPGA under cardinality and knapsack constraints.

We consider the coverage function in the SCM setting, defined as $F(x) = \sum_{i=1}^{n} \phi_i(m_i(x))$, where $m_i(x)$ denotes the linear function representing the coverage contribution of component $i$ in $x$, and $\phi_i(x) = \min\{1, x\}$ is a concave function. Thus, $F(x)$ is the continuous extension of the coverage function. We compare AAPGA and PGA under both the cardinality constraint $\{\mathbf{x} \in \{0,1\}^n \mid \sum_{i=1}^{n} x_i \leq k\}$ and the knapsack constraint $\{\mathbf{x} \in [0,1]^n \mid \sum_{i=1}^{n} c_i x_i \leq B\}$.

Figure 2 illustrates the convergence behavior of PGA and AAPGA across four representative graph datasets: CA-GrQc, Wikipedia Vote, EU Email, and Facebook Ego Network. The datasets are drawn from the SNAP repository: Facebook Ego Network (348 vertices, 5038 edges), EU Email (1005 vertices, 25,571 edges), CA-GrQc (5242 vertices, 28,980 edges), and Wikipedia Vote (8298 vertices, 103,689 edges).

Across all eight settings (two constraints per dataset), AAPGA consistently outperforms PGA in both convergence speed and final objective value. The advantage is particularly pronounced in the early iterations: for instance, under the knapsack constraint on the EU Email dataset, AAPGA achieves an objective value of 748.7 within just 20 iterations, whereas PGA only reaches 743.8 after 50 iterations. Likewise, on the Wikipedia Vote dataset, PGA converges to 4600.6, while AAPGA attains a higher value of 4629.0. These results demonstrate that AAPGA not only converges significantly faster in the early stages but also achieves superior final objective values compared to PGA.

In classical PGA, the step size must be carefully tuned to avoid oscillations, particularly when the objective exhibits steep curvature in certain directions. This cautious choice often slows progress along flatter directions, resulting in overall slow convergence. In contrast, AAPGA employs a more efficient step size strategy that mitigates oscillations caused by overly aggressive fixed steps while simultaneously accelerating progress along low-curvature directions. This explains both the rapid initial gains and the consistently higher function values observed across datasets.

## 4.2 EXPERIMENTS ON SMALL-SCALE SUBMODULAR MAXIMIZATION

In many applications, submodular optimization is often studied under small-scale settings. We focus on the problem of submodular maximization under two types of constraints: cardinality and knapsack. The objective function is defined as the probabilistic coverage problem. Let the ground set be $V = \{1, 2, \ldots, n\}$. Each candidate element $i \in S \subseteq V$ covers a target node $j \in V$ with probability $p_{ij} \in [0, 1]$. The function is given by $f(S) = \sum_{j \in V} \left(1 - \prod_{i \in S}(1 - p_{ij})\right)$. In the SCM form, the function is written as $f(x) = \sum_{j \in V} \left(1 - \exp\left(\sum_{i \in V} x_i \log(1 - p_{ij})\right)\right)$. Throughout, we assume an undirected graph $G = (V, E)$ and define the coverage probabilities by $p_{ij} = 0.8$ if $(i, j) \in E$ and $p_{ij} = 0$ otherwise, where we further take $p_{ij} = p_{ji}$ for undirected edges and set $p_{ii} = 0$.

We evaluate our methods on two synthetic graphs, namely the Stochastic Block Model (2000 nodes, 56,150 edges) and the Erdős–Rényi random graph (4000 nodes, 40,031 edges), as well as two real-world graphs: the EU Email network (1005 vertices, 25,571 edges) and the Wikipedia Vote network (8298 vertices, 103,689 edges). On these four datasets, we compare the performance of different

Table 2: Probabilistic coverage maximization under a cardinality constraint. Comparison of algorithms on four datasets; for each dataset, the left entry reports the objective value (higher is better) and the right entry reports wall-clock time in seconds (lower is better). In each column, the best, second-best, and third-best results are marked with $\star$, $\dagger$, and $\ddagger$, respectively.

| Algorithm | EU-Email | | Facebook-ego | | Stochastic Block Model | | Erdős–Rényi | |
|---|---|---|---|---|---|---|---|---|
| | Result | Time | Result | Time | Result | Time | Result | Time |
| FAST | 636.67 | 0.11 | 1494.33 | 1.16 | 1355.20 | 0.03 | 3491.66 | 0.27 |
| LS+PGB | 781.55 | 0.09 | 3145.43 | 1.06 | 1400.58 | 0.11 | 3541.94 | 0.34 |
| Stream | 784.74 | 0.99 | 3441.45 | 25.47 | 1440.89 | 1.90 | 3538.56 | 6.88 |
| Greedy | 864.61$^\star$ | 4.96 | 3676.31$^\star$ | 145.74 | 1654.00$^\star$ | 9.73 | 3773.46$^\star$ | 42.02 |
| Lazier-greedy | 853.75 | 1.02 | 3610.24$^\ddagger$ | 29.18 | 1628.71$^\ddagger$ | 1.98 | 3748.74$^\dagger$ | 8.44 |
| PGA+Rounding | 862.69$^\ddagger$ | 1.68 | 1661.63 | 1.49 | 1427.04 | 2.09 | 3652.71 | 2.50 |
| **AAPGA+Rounding** | **864.60$^\dagger$** | **1.12** | **3653.14$^\dagger$** | **1.84** | **1642.38$^\dagger$** | **1.78** | **3714.83$^\ddagger$** | **1.30** |

Table 3: Probabilistic coverage maximization under the knapsack constraint. For each dataset, the left entry reports the objective value (higher is better) and the right entry reports wall-clock time in seconds (lower is better).

| Algorithm | EU-Email | | Facebook-ego | | SBM | | Erdős–Rényi | |
|---|---|---|---|---|---|---|---|---|
| | Result | Time | Result | Time | Result | Time | Result | Time |
| Stream | 826.82 | 2.08 | 4629.69 | 37.88 | 1986.84 | 2.00 | 4023.25 | 9.02 |
| Lazier-greedy | 879.72 | 1.18 | 4885.69 | 34.51 | 1999.92 | 1.12 | 4208.90 | 7.62 |
| **AAPGA+Rounding** | **874.89** | **0.86** | **4850.83** | **3.55** | **1999.72** | **0.48** | **4183.47** | **3.38** |

algorithms under both the cardinality and knapsack constraints. Due to the use of stochastic algorithms, the results and runtime of all algorithms are averaged over 50 runs.

For the cardinality-constrained setting, we compare against several well-established algorithms, including the classical serial methods GREEDY (Nemhauser et al., 1978) and LAZIER-GREEDY (Mirzasoleiman et al., 2015), the parallel implementations FAST (Breuer et al., 2020), LS+PGB (Chen et al., 2021), streaming methods (Badanidiyuru et al., 2014), and continuous relaxations with PGA followed by pipage rounding (Bai et al., 2018). Since SCM functions can be efficiently computed on GPUs, we apply GPU acceleration to all methods, regardless of whether they are originally designed as GPU algorithms. We also evaluate our proposed AAPGA combined with pipage rounding (AAPGA+Rounding) under the same constraint. Throughout, we set the cardinality budget to $k = 50$; that is, each method selects 50 elements to maximize the submodular objective.

For the knapsack-constrained setting, we use as baselines the $\frac{1}{2}(1 - 1/e)$-approximation greedy algorithm Khuller et al. (1999), which we implement with lazy updates to accelerate computation, as well as the streaming methods of Huang et al. (2020). In addition, we evaluate our proposed AAPGA combined with randomized rounding (AAPGA+Rounding). The knapsack budget is defined as $B \sim \text{Unif}(\sum_{i=1}^n c_i/50, \sum_{i=1}^n c_i/20)$, that is, $B$ is drawn uniformly at random from the interval between $\sum_{i=1}^n c_i/50$ and $\sum_{i=1}^n c_i/20$.

We evaluate and compare the solution quality and running time of all algorithms on the four graphs, with the results summarized in Tables 2 and 3.

Unless otherwise specified, AAPGA is run for 100 iterations, while PGA is run for 200 iterations. For AAPGA, $L_0 = 1, \beta = 2$. Under the cardinality constraint, AAPGA+Rounding consistently ranks among the top three in objective value across datasets, while providing substantial speedups. For example, on the mid-scale Erdős–Rényi random graph, AAPGA+Rounding is $32.3\times$ faster than Greedy. Because Greedy and Lazier-Greedy are difficult to parallelize effectively, they exhibit significantly higher running times. In contrast, methods that leverage GPU parallelism consistently achieve markedly lower wall-clock time, underscoring the benefits of GPU acceleration for submodular maximization at this scale.

Under the knapsack constraint, our method consistently outperforms the streaming baseline in terms of objective value. Compared to Lazier-greedy, it sacrifices a small amount of solution quality

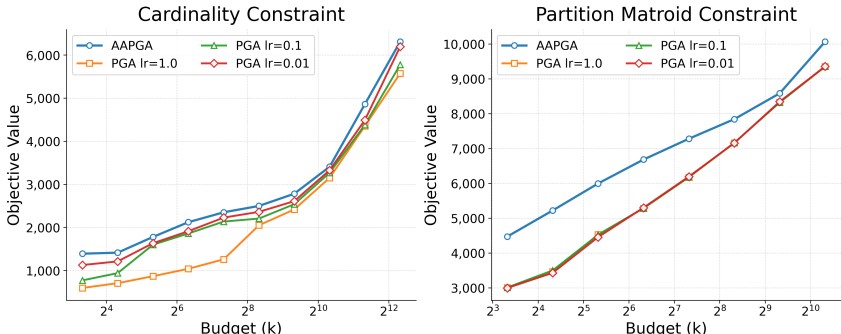

Figure 3: Comparison of AAPGA with pipage rounding and PGA with pipage rounding under cardinality and partition matroid constraints.

but achieves a dramatic speedup. For instance, on the *Facebook-ego* graph, AAPGA+Rounding is nearly $9.72\times$ faster than Lazier-greedy, representing a substantial improvement in computational efficiency.

### 4.3 EXPERIMENTS ON LARGE-SCALE SUBMODULAR MAXIMIZATION

Large-scale submodular maximization is particularly challenging due to its high memory requirements and the necessity of multi-GPU parallelization. In this experiment, we focus on comparing PGA and AAPGA (instead of including other baselines) in order to clearly highlight the benefits brought by acceleration. The dataset under consideration is the Google+ social network, collected from users who voluntarily shared their circles via the "share circle" feature. The submodular function is defined as the number of users covered by selecting $k$ seed users (Singer, 2012). The dataset consists of 107,614 vertices and 30,494,866 edges. The cardinality budget $k$ ranges from 10 to 5120, increasing exponentially.

As shown in Figure 3, the objective value increases monotonically under both cardinality and partition matroid constraints. More importantly, AAPGA consistently exhibits a faster growth rate than PGA across all learning-rate settings. Since the cardinality constraint is simpler than the partition matroid constraint, PGA combined with pipage rounding performs comparatively better under the cardinality setting. Overall, these results demonstrate that AAPGA, when integrated with pipage rounding, provides a more effective and scalable solution for large-scale submodular maximization.

### ACKNOWLEDGMENTS

This work was partially supported by National Natural Science Foundation of China (No. 12131003) and Beijing Natural Science Foundation (No. Z220004).

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

APPENDIX A: PROOF OF THEOREM 1

For convenience in the proof, we define the exact projection

$$p_L(\mathbf{y}^{(t)}) \ = \ \Pi_{\mathcal{P}}\Big(\mathbf{y}^{(t)} + \tfrac{1}{L}\gamma_F(\mathbf{y}^{(t)})\Big).$$

Since $F$ is defined over $\mathbb{R}^n$, the maximization problem

$$\max_{\mathbf{x}\in\mathcal{P}} F(\mathbf{x})$$

can be equivalently reformulated as the minimization problem

$$\min_{\mathbf{x}\in\mathbb{R}^n} h(\mathbf{x}) = \min_{\mathbf{x}\in\mathbb{R}^n}\big(-F(\mathbf{x}) - \mathbb{1}_{\mathcal{P}}(\mathbf{x})\big) = \min_{\mathbf{x}\in\mathbb{R}^n}\big(-F(\mathbf{x}) - g(\mathbf{x})\big),$$

where $h(\mathbf{x}) := -F(\mathbf{x}) - \mathbb{1}_{\mathcal{P}}(\mathbf{x})$ and $g(\mathbf{x}) := \mathbb{1}_{\mathcal{P}}(\mathbf{x})$.

**Lemma 2.** *For any $\mathbf{y} \in \mathbb{R}^n$, if $\gamma_g(\mathbf{y}) \in \partial g(\mathbf{z})$, then with $\mathbf{z}' = p'_L(\mathbf{y})$ and $\mathbf{z} = p_L(\mathbf{y})$, the following holds:*

$$-\nabla F(\mathbf{y}) + L(\mathbf{z} - \mathbf{y}) - \gamma(\mathbf{y}) \ = \ L\big(\mathbf{z}' - \mathbf{z}\big).$$

*Proof.* The result follows directly from the optimality conditions of the strongly convex function $Q_L$. Specifically, we have

$$-\nabla F(\mathbf{y}) + L(p_L(\mathbf{y}) - \mathbf{y}) - \gamma(\mathbf{y}) = 0,$$

which further implies

$$-\nabla F(\mathbf{y}) + L\big(p'_L(\mathbf{y}) - \mathbf{y}\big) - \gamma(\mathbf{y}) = L\big(p'_L(\mathbf{y}) - p_L(\mathbf{y})\big).$$

$\square$

**Lemma 3.** *Let $\mathbf{y} \in \mathbb{R}^n$ and $L > 0$. Suppose that*

$$F(p'_L(\mathbf{y})) \geq Q(p'_L(\mathbf{y}), \mathbf{y}).$$

*For any $\mathbf{x} \in [0, 1]^n$, it holds that*

$$h(\mathbf{x}) - h(p'_L(\mathbf{y})) \ \geq \ \tfrac{L}{2}\|p'_L(\mathbf{y}) - \mathbf{y}\|^2 + L\langle \mathbf{y} - \mathbf{x}, p'_L(\mathbf{y}) - \mathbf{y}\rangle - 2LR\delta.$$

*Proof.* From the given conditions, we can derive

$$h(\mathbf{x}) - h(p'_L(\mathbf{y})) = h(\mathbf{x}) + F(p'_L(\mathbf{y})) \ \geq \ h(\mathbf{x}) + Q(p'_L(\mathbf{y}), \mathbf{y}). \tag{10}$$

By the concavity of $F$ and $h$, we have

$$-F(\mathbf{x}) \ \geq \ -F(\mathbf{y}) - \langle \mathbf{x} - \mathbf{y}, \nabla F(\mathbf{y})\rangle, \tag{11}$$

$$-g(\mathbf{x}) \ \geq \ -g(p'_L(\mathbf{y})) - \langle \mathbf{x} - p'_L(\mathbf{y}), \gamma_g(\mathbf{y})\rangle. \tag{12}$$

Adding the two inequalities equation 11 and equation 12 yields

$$h(\mathbf{x}) \ \geq \ -F(\mathbf{y}) - \langle \mathbf{x} - \mathbf{y}, \nabla F(\mathbf{y})\rangle - g(p'_L(\mathbf{y})) - \langle \mathbf{x} - p'_L(\mathbf{y}), \gamma_g(\mathbf{y})\rangle. \tag{13}$$

By definition,

$$Q(p'_L(\mathbf{y}), \mathbf{y}) = F(\mathbf{y}) + \langle p'_L(\mathbf{y}) - \mathbf{y}, \nabla F(\mathbf{y})\rangle - \tfrac{L}{2}\|p'_L(\mathbf{y}) - \mathbf{y}\|^2. \tag{14}$$

Substituting equation 13 and equation 14 into equation 10, we obtain

$$
\begin{aligned}
h(\mathbf{x}) - h(p'_L(\mathbf{y})) \ &\geq \ -\langle \mathbf{x} - p'_L(\mathbf{y}), \nabla F(\mathbf{y}) + \gamma_g(\mathbf{y})\rangle - \tfrac{L}{2}\|p'_L(\mathbf{y}) - \mathbf{y}\|^2. \\
&= L\langle \mathbf{x} - p'_L(\mathbf{y}), \mathbf{y} - p'_L(\mathbf{y})\rangle - \tfrac{L}{2}\|p'_L(\mathbf{y}) - \mathbf{y}\|^2 - L\langle \mathbf{x} - p'_L(y), p_L(\mathbf{y}) - p'_L(\mathbf{y})\rangle \\
&= \tfrac{L}{2}\|p'_L(\mathbf{y}) - \mathbf{y}\|^2 + L\langle \mathbf{y} - \mathbf{x}, p'_L(\mathbf{y}) - \mathbf{y}\rangle - L\langle \mathbf{x} - p'_L(\mathbf{y}), p'_L(\mathbf{y}) - p_L(\mathbf{y})\rangle \\
&\geq \tfrac{L}{2}\|p'_L(\mathbf{y}) - \mathbf{y}\|^2 + L\langle \mathbf{y} - \mathbf{x}, p'_L(\mathbf{y}) - \mathbf{y}\rangle - 2LR\delta.
\end{aligned}
$$

$\square$

**Lemma 4.** *For the sequence $\{\mathbf{x}^{(t)}, \mathbf{y}^{(t)}\}$ generated by Accelerated Approximate Projected Gradient Ascent, for all $t \geq 1$ we have*

$$2L_t^{-1}\alpha_t^2 \mathbf{v}^{(t)} - 2L_{t+1}^{-1}\alpha_{t+1}^2 \mathbf{v}^{(t+1)} \geq \|\mathbf{u}^{(t+1)}\|^2 - \|\mathbf{u}^{(t)}\|^2 - 4R\delta\,\alpha_{t+1}^2,$$

*where $v^{(t)} := h(\mathbf{x}^{(t)}) - h(\mathbf{x}^*)$ and $\mathbf{u}^{(t)} := \alpha_t \mathbf{x}^{(t)} - (\alpha_t - 1)\mathbf{x}^{(t-1)} - \mathbf{x}^*$.*

*Proof.* By substituting $\mathbf{x} = \mathbf{x}^{(t)}$ and $\mathbf{y} = \mathbf{y}^{(t+1)}$ into Lemma 3, or alternatively $\mathbf{x} = \mathbf{x}^*$ and $\mathbf{y} = \mathbf{y}^{(t+1)}$, we obtain

$$2L_{t+1}^{-1}\big(h(\mathbf{x}^{(t)}) - h(\mathbf{y}^{(t+1)})\big) \geq \|\mathbf{x}^{(t+1)} - \mathbf{y}^{(t+1)}\|^2 + 2\langle \mathbf{x}^{(t+1)} - \mathbf{y}^{(t+1)},\, \mathbf{y}^{(t+1)} - \mathbf{x}^{(t)}\rangle - 4R\delta. \quad (15)$$

and

$$-2L_{t+1}^{-1}v^{(t+1)} \geq \|\mathbf{x}^{(t+1)} - \mathbf{y}^{(t+1)}\|^2 + 2\langle \mathbf{x}^{(t+1)} - \mathbf{y}^{(t+1)},\, \mathbf{y}^{(t+1)} - \mathbf{x}^*\rangle - 4R\delta. \quad (16)$$

Multiplying the inequality equation 15 by $(\alpha_{t+1} - 1)$ and adding it to the inequality equation 16 gives

$$2L_{t+1}^{-1}\Big[(\alpha_{t+1} - 1)v^{(t)} - \alpha_{t+1}v^{(t+1)}\Big] \geq \alpha_{t+1}\|\mathbf{x}^{(t+1)} - \mathbf{y}^{(t+1)}\|^2$$

$$+ 2\Big\langle \mathbf{x}^{(t+1)} - \mathbf{y}^{(t+1)},\, \alpha_{t+1}\mathbf{y}^{(t+1)} - (\alpha_{t+1} - 1)\mathbf{x}^{(t)} - \mathbf{x}^* \Big\rangle$$

$$- 4R\delta\,\alpha_{t+1}.$$

Multiplying both sides by $\alpha_{t+1}$ and using the identity $\alpha_t^2 = \alpha_{t+1}^2 - \alpha_{t+1}$, we obtain

$$2L_{t+1}^{-1}\Big(\alpha_t^2 \mathbf{v}^{(t)} - \alpha_{t+1}^2 \mathbf{v}^{(t+1)}\Big) \geq \|\alpha_{t+1}(\mathbf{x}^{(t+1)} - \mathbf{y}^{(t+1)})\|^2$$

$$+ 2\alpha_{t+1}\Big\langle \mathbf{x}^{(t+1)} - \mathbf{y}^{(t+1)},\, \alpha_{t+1}\mathbf{y}^{(t+1)} - (\alpha_{t+1} - 1)\mathbf{x}^{(t)} - \mathbf{x}^* \Big\rangle \qquad (17)$$

$$- 4R\delta\,\alpha_{t+1}^2$$

$$\geq \big\|\alpha_{t+1}\mathbf{x}^{(t+1)} - (\alpha_{t+1} - 1)\mathbf{x}^{(t)} - \mathbf{x}^*\big\|^2$$

$$- \big\|\alpha_{t+1}\mathbf{y}^{(t+1)} - (\alpha_{t+1} - 1)\mathbf{x}^{(t)} - \mathbf{x}^*\big\|^2 - 4R\delta\,\alpha_{t+1}^2.$$

The last step follows from the identity

$$\|\mathbf{b} - \mathbf{a}\|^2 + 2\langle \mathbf{b} - \mathbf{a},\, \mathbf{a} - \mathbf{c}\rangle = \|\mathbf{b} - \mathbf{c}\|^2 - \|\mathbf{a} - \mathbf{c}\|^2.$$

In addition, we have the identity

$$\alpha_{t+1}\mathbf{y}^{(t)} = \alpha_{t+1}\mathbf{x}^{(t)} + (\alpha_t - 1)\left(\mathbf{x}^{(t)} - \mathbf{x}^{(t-1)}\right).$$

Substituting the definition of $\mathbf{u}^{(t)}$ into the inequality yields

$$2L_{t+1}^{-1}\Big(\alpha_t^2 v^{(t)} - \alpha_{t+1}^2 v^{(t+1)}\Big) \geq \|\mathbf{u}^{(t+1)}\|^2 - \|\mathbf{u}^{(t)}\|^2 - 4R\delta\,\alpha_{t+1}^2.$$

Finally, since $L_{t+1} \geq L_t$, we conclude

$$2L_t^{-1}\alpha_t^2 v^{(t)} - 2L_{t+1}^{-1}\alpha_{t+1}^2 v^{(t+1)} \geq \|\mathbf{u}^{(t+1)}\|^2 - \|\mathbf{u}^{(t)}\|^2 - 4R\delta\,\alpha_{t+1}^2.$$

$\square$

We now conclude with the proof of Theorem 1.

*Proof.* It follows from Lemma 3 that

$$
h(\mathbf{x}^{(1)}) - h(\mathbf{x}^*) = h(p'_{L_1}(\mathbf{y}^{(1)})) - h(\mathbf{x}^*)
$$

$$
\leq -\tfrac{L_1}{2} \|p'_{L_1}(\mathbf{y}^{(1)}) - \mathbf{y}^{(1)}\|^2 - L_1\langle \mathbf{y}^{(1)} - \mathbf{x}^*, \, p'_{L_1}(\mathbf{y}^{(1)}) - \mathbf{y}^{(1)} \rangle + 2L_1 R\delta
$$

$$
= -\tfrac{L_1}{2} \|\mathbf{x}^{(1)} - \mathbf{y}^{(1)}\|^2 - L_1\langle \mathbf{y}^{(1)} - \mathbf{x}^*, \, \mathbf{x}^{(1)} - \mathbf{y}^{(1)} \rangle + 2L_1 R\delta
$$

$$
= -\tfrac{L_1}{2} \left( \|\mathbf{x}^{(1)} - \mathbf{x}^*\|^2 - \|\mathbf{y}^{(1)} - \mathbf{x}^*\|^2 \right) + 2L_1 R\delta. \tag{18}
$$

By summing Lemma 4 over $t = 1$ to $T - 1$, we obtain

$$
2L_1^{-1}\alpha_1^2 v^{(1)} - 2L_T^{-1}\alpha_T^2 v^{(T)} \geq \|\mathbf{u}_T\|^2 - \|\mathbf{u}_1\|^2 - \sum_{i=2}^{T} 4R\delta t_i^2.
$$

Since $\alpha_1 = 1$ and the term $\|\mathbf{u}_T\|^2$ can be discarded, we have

$$
v^{(T)} = h(\mathbf{x}^{(T)}) - h(\mathbf{x}^*) \leq \frac{L_T}{L_1\alpha_T^2} \left( h(\mathbf{x}^{(1)}) - h(\mathbf{x}^*) \right) + \frac{L_T}{2\alpha_T^2} \|\mathbf{x}^{(1)} - \mathbf{x}^*\|^2 + \frac{2R\delta L_T \sum_{i=2}^{T} \alpha_i^2}{\alpha_T^2}. \tag{19}
$$

Since $t_i \leq i$, combining equation 18 and equation 19 yields the desired bound. Here we omit the proof of the inequality $(t+1)/2 \leq \alpha_t \leq t$.

$$
h(\mathbf{x}^{(T)}) - h(\mathbf{x}^*) \leq \tfrac{2L_T}{(T+1)^2} \|y_1 - x^*\|^2 + \frac{2R\delta L_T \left( \sum_{i=2}^{T} t^2 + 1 \right)}{(T+1)^2/4}
$$

$$
\leq \tfrac{L_T}{2(T+1)^2} \|y_1 - x^*\|^2 + 4R\delta L_T \cdot \frac{T(T+1)(2T+1)}{3(T+1)^2} \tag{20}
$$

$$
= \tfrac{L_T}{2(T+1)^2} \|y_1 - x^*\|^2 + \frac{4R\delta L_T}{3} \cdot \frac{T(2T+1)}{T+1}
$$

$$
= O(n/T^2) + R\delta L_T O(T).
$$

Recalling that $h(\mathbf{x}) := -F(\mathbf{x}) - \mathbb{1}_{\mathcal{P}}(\mathbf{x})$, we obtain

$$
F(\mathbf{x}^*) - F(\mathbf{x}^{(T)}) \leq O(n/T^2) + R\delta L_T O(T). \tag{21}
$$

In particular, when $\delta = O(1/T^3)$, the algorithm achieves a convergence rate of $O(n/T^2)$.

$\square$

## APPENDIX B: PROJECTION ALGORITHMS

In this section, we employ projection methods to compute approximate projections for cardinality, knapsack, and matroid constraints in $O(n \log(n/\delta))$ time.

### PROJECTIONS UNDER CARDINALITY AND KNAPSACK CONSTRAINTS

Consider a knapsack constraint defined by item sizes $\mathbf{c} = (c_1, c_2, \ldots, c_n)$ and knapsack capacity $B$. The cardinality constraint can be regarded as a special case with $\mathbf{c} = \mathbf{1}_n$ and capacity $k$. The corresponding feasible polytope is

$$
\mathcal{P} = \{ \mathbf{x} \in \mathbb{R}^n \mid \mathbf{c} \cdot \mathbf{x} \leq B, \, \mathbf{0} \leq \mathbf{x} \leq \mathbf{1} \}.
$$

For any given point $\mathbf{v}$, the projection problem can be formulated as

$$
\min_{\mathbf{w}} \tfrac{1}{2}\|\mathbf{w} - \mathbf{v}\|_2^2 \quad \text{s.t.} \sum_{i=1}^{n} c_i w_i \leq B, \; 0 \leq w_i \leq 1 \quad \forall i \in [n].
$$

The corresponding Lagrangian can be written as

$$\mathcal{L}(\mathbf{w}, \alpha, \boldsymbol{\beta}, \boldsymbol{\gamma}) = \tfrac{1}{2}\|\mathbf{w} - \mathbf{v}\|_2^2 + \alpha\left(\sum_{i=1}^{n} c_i w_i - B\right) - \boldsymbol{\beta}^\top \mathbf{w} + \boldsymbol{\gamma}^\top(\mathbf{w} - \mathbf{1}_n),$$

where $\alpha \in \mathbb{R}$, $\boldsymbol{\beta}, \boldsymbol{\gamma} \in \mathbb{R}_+^n$ are dual variables.

Taking the derivative with respect to $w_i$ yields

$$\frac{\partial L}{\partial w_i} = w_i - v_i + \alpha c_i - \beta_i + \gamma_i = 0.$$

By the KKT conditions, we have

$$\beta_i w_i = 0, \quad \gamma_i(w_i - 1) = 0, \quad \forall i \in [n].$$

It follows that the solution takes the form

$$\mathbf{w}(\alpha) = \min\{\mathbf{1}, \max\{\mathbf{0}, \mathbf{v} - \alpha \cdot \mathbf{c}\}\}.$$

If the optimal solution lies in the interior with $\sum_{i=1}^{n} c_i w_i < B$, then $\alpha = 0$ and the projection is simply $\mathbf{v}$. Otherwise, when the solution lies on the boundary with $\sum_{i=1}^{n} c_i w_i < B$, an appropriate $\alpha > 0$ must be chosen such that

$$\sum_{i=1}^{n} \mathbf{c} \cdot \mathbf{w}(\alpha) = B.$$

Since $\mathbf{c} \cdot \mathbf{w}(\alpha)$ is monotone in $\alpha$, we can efficiently determine the optimal $\alpha$ using binary search. The initial search interval is set to $[0, \max_{i \in [n]} v_i/c_i]$. Performing bisection until the error is within $\delta/\sqrt{n}$ requires $O(\log(n/\delta))$ iterations. Hence, the overall projection can be computed in $O(n\log(n/\delta))$ time.

---

**Algorithm 3** Projection onto the Knapsack Polytope via Bisection

---

**Require:** $\mathbf{v} \in \mathbb{R}^n$, item sizes $\mathbf{c} = (c_1, c_2, \ldots, c_n)$, knapsack capacity $B$, tolerance $\delta$
1: **if** $\mathbf{c} \cdot \mathbf{v} \le B$ **then**
2:     **return** $\mathbf{v}$
3: **end if**
4: Initialize $\mathbf{z} \leftarrow \mathbf{v}$
5: $lo \leftarrow 0, \quad hi \leftarrow \max_{i \in [n]} v_i/c_i$
6: **while** $hi - lo > \delta/\sqrt{n}$ **do**
7:     $\alpha \leftarrow (lo + hi)/2$
8:     $s \leftarrow \sum_{i=1}^{n} \min\{1, \max\{0, v_i - \alpha c_i\}\}$
9:     **if** $s > B$ **then**
10:         $lo \leftarrow \alpha$
11:     **else**
12:         $hi \leftarrow \alpha$
13:     **end if**
14: **end while**
15: $\mathbf{z} \leftarrow \min\{\mathbf{1}, \max\{\mathbf{0}, \mathbf{v} - hi \cdot \mathbf{c}\}\}$
16: **return** $\mathbf{z}$

---

PROJECTIONS UNDER PARTITION MATROIDS

For a partition matroid, the feasible region can be written as

$$\mathcal{P} = \left\{ x \in [0,1]^n \;\middle|\; \sum_{i \in G_j} x_i \le r_j, \; \forall j \in [m] \right\},$$

where $\{G_1, G_2, \ldots, G_m\}$ is a partition of the ground set. Hence, projection reduces to solving a separate projection subproblem for each block $G_j$. Each block requires $O(|G_j|\log(n/\delta))$ time, and summing over all blocks yields the total complexity $O(n\log(n/\delta))$.

APPENDIX C: PROOF OF LEMMA 1

**Lemma 5.** *(Bounds in the tails of the distribution of a weighted sum of independent Bernoulli trials) Let $a_1, a_2, \ldots, a_r$ be reals in $(0, 1]$. Let $X_1, X_2, \ldots, X_n$ be independent Bernoulli trials with $E[X_j] = p_j$. For the random variable $\Psi = \sum_{j=1}^{n} a_j X_j$ and for $\delta \in (0, 1]$,*

$$\Pr[\Psi < (1 - \eta)E\Psi] < e^{\frac{-\eta^2 E\psi}{2}}$$

We now turn to the proof of Lemma 1.

We begin by proving that $F_m(x) \leq F(x)$. We prove the result by induction on the dimension $n$.

**Base case:** When $n = 1$, we have

$$F_m(x) = xf(1) + (1 - x)f(0) = xF(1) + (1 - x)F(0),$$

where $F(0)$ and $F(1)$ are SCM evaluations at the endpoints. Since $\phi_i$ are concave functions and $m_i(x)$ is linear in $x$, Jensen's inequality gives

$$xF(1) + (1 - x)F(0) \leq F(x) = \sum_i \phi_i(m_i(x)).$$

**Inductive step:** Assume the statement holds for dimension $n = k$. For $\mathbf{x} \in [0, 1]^k$, define:

$$F_m(\mathbf{x}) = x_k F_m(\mathbf{e}_k \vee \mathbf{x}) + (1 - x_k)F_m(\bar{\mathbf{e}}_k \wedge \mathbf{x}),$$

where $\mathbf{e}_k$ denotes the $k$-th standard basis vector, and $\bar{\mathbf{e}}_k$ denotes the vector with 0 in the $k$-th coordinate and 1 in all other coordinates. By the inductive hypothesis, we have

$$F_m(\mathbf{e}_k \vee \mathbf{x}) \leq F(\mathbf{e}_k \vee \mathbf{x}), \quad F_m(\bar{\mathbf{e}}_k \wedge \mathbf{x}) \leq F(\bar{\mathbf{e}}_k \wedge \mathbf{x}).$$

Using the convex combination and the concavity of $f$, we get:

$$F_m(\mathbf{x}) \leq x_k F(\mathbf{e}_k \vee \mathbf{x}) + (1 - x_k)F(\bar{\mathbf{e}}_k \wedge \mathbf{x}) \leq F(\mathbf{x}).$$

Hence, by induction, the inequality $F(\mathbf{x}) \geq F_m(\mathbf{x})$ holds for all $\mathbf{x} \in [0, 1]^n$.

The following proof is more intricate. Recall that the multilinear extension is defined as $F_m(\mathbf{x}) = \mathbb{E}_{A \sim D_{\mathbf{x}}}[f(A)]$. One natural approach is to estimate the probability $\Pr[f(A) \geq CF(\mathbf{x})]$, which immediately yields the lower bound:

$$F_m(\mathbf{x}) \geq \Pr[f(A) \geq CF(\mathbf{x})] \cdot CF(\mathbf{x}).$$

We begin by analyzing the function $m_1$. According to Lemma 5, we have:

$$\Pr[m_1(A) < (1 - \eta)\mathbb{E}_{A \sim D_{\mathbf{x}}}[m_1(A)]] < \exp\left(-\frac{\eta^2 \mathbb{E}_{A \sim D_{\mathbf{x}}}[m_1(A)]}{2 \max(m_1)}\right).$$

Since $\mathbb{E}_{A \sim D_x}[m_1(A)] = m_1 \cdot x$, it follows that:

$$\Pr[m_1(A) < (1 - \eta)m_1 \cdot x] < \exp\left(-\frac{\eta^2 m_1 \cdot x}{2 \max(m_1)}\right).$$

Define the event $\overline{B}_{m_1}(\eta) = \{m_1(A) < (1 - \eta)\mathbb{E}_{A \sim D_x}[m_1(A)]\}$. Then we have:

$$\Pr\left[\overline{B}_{m_1}(\eta)\right] < \exp\left(-\frac{\eta^2 m_{v_1} \cdot x}{2 \max(m_1)}\right).$$

If the event $B_{m_1}(\eta)$ occurs, that is, $m_1(A) \geq (1 - \eta)\mathbb{E}_{A \sim D_x}[m_1(A)]$, then by the monotonicity of the function $\phi$, we obtain: $\phi(m_1(A)) \geq \phi((1 - \eta)\mathbb{E}_{A \sim D_x}[m_1(A)])$. Using the concavity of $\phi$, it follows that:

$$\phi((1 - \eta)\mathbb{E}_{A \sim D_x}[m_1(A)]) \geq (1 - \eta)\phi(\mathbb{E}_{A \sim D_x}[m_1(A)]) \geq (1 - \eta)\mathbb{E}_{A \sim D_x}[\phi(m_1(A))].$$

Hence,

$$\Pr\left[\phi(m_1(A)) \geq (1 - \eta)\mathbb{E}_{A \sim D_x}[\phi(m_1(A))]\right] \geq 1 - \exp\left(-\frac{\eta^2 m_{v_1} \cdot x}{2\max(m_1)}\right).$$

Regardless of whether the events $\overline{B}_{m_i}(\eta)$ are dependent or not, a union bound gives:

$$\Pr[\overline{B}_F(\eta)] \leq \sum_{i \in V} \Pr[\overline{B}_{m_i}(\eta)].$$

Therefore, we obtain:

$$\Pr\left[f(A) = \sum_{i \in V} \phi(m_i(A)) \geq \sum_{i \in V}(1 - \eta)\mathbb{E}_{A \sim D_x}[\phi(m_i(A))]\right]$$
$$= \Pr\left[f(A) \geq (1 - \eta)F_m(x)\right]$$
$$\geq 1 - |V| \cdot \max_{i \in V}\left\{\exp\left(-\frac{\eta^2 m_i \cdot x}{2\max(m_i)}\right)\right\}.$$

Combining this with the earlier bound yields:

$$F(x) \geq (1 - \eta)\left(1 - |V| \cdot \max_{i \in V}\left\{\exp\left(-\frac{\eta^2 m_i \cdot x}{2\max(m_i)}\right)\right\}\right)F_m(x).$$

## APPENDIX D: PROOF OF THEOREM 2

We first analyze the SCM maximization problem under cardinality and partition matroid constraints. Let $F$ denote the concave extension of $f$. By applying the AAPGA algorithm to compute $\hat{\mathbf{x}}$, we obtain

$$F(\hat{\mathbf{x}}) \geq (1 - \varepsilon)\max_{\mathbf{x} \in \mathcal{P}} F(\mathbf{x}).$$

According to Lemma 1, the following inequality holds:

$$F_m(\hat{\mathbf{x}}) \geq \max_{\eta \in [0,1]}(1 - \eta)\left[1 - |V|\exp\left(-\eta^2 \triangle(\hat{\mathbf{x}})\right)\right]F(\hat{\mathbf{x}}),$$

where

$$\triangle(\hat{\mathbf{x}}) = \min_{i \in V}\frac{m_i \cdot \hat{\mathbf{x}}}{\max m_i} \geq \min_{i \in V}\frac{\min m_i \cdot k}{\max m_i}.$$

$m_i \cdot \hat{\mathbf{x}} \geq \min m_i \cdot k$ holds because $\hat{\mathbf{x}}$ lies on the face of $\mathcal{P}$.

Therefore, we can further derive the following bound:

$$F_m(\hat{\mathbf{x}}) \geq \max_{\eta \in [0,1]}(1 - \eta)\left[1 - |V|\exp\left(-\eta^2 \min_{i \in V}\frac{\min m_i \cdot k}{\max m_i}\right)\right]F(\hat{\mathbf{x}}).$$

Hence,

$$F_m(\hat{\mathbf{x}}) \geq (1 - \varepsilon)\max_{\mathbf{x} \in \mathcal{P}} F(\mathbf{x}) \geq \max_{\eta \in [0,1]}(1 - \varepsilon)(1 - \eta)\left[1 - |V|\exp\left(-\eta^2 \min_{i \in V}\frac{\min m_i \cdot k}{\max m_i}\right)\right]F(\hat{\mathbf{x}}). \tag{22}$$

By relaxing the expression, we obtain

$$(1 - \eta)\left[1 - |V|\exp\left(-\eta^2 \min_{i \in V}\frac{\min m_i}{\max m_i}\right)\right] \geq 1 - \eta - e^{-\Omega(\eta^2)}.$$

Since pipage rounding guarantees that the integral solution is no worse than the multilinear relaxation, the rounded solution $\mathbf{x}'$ satisfies

$$\mathbb{E}[f(\mathbf{x}')] \geq F_m(\hat{\mathbf{x}}).$$

Therefore, we conclude that

$$\mathbb{E}[f(\mathbf{x}')] \geq (1 - \varepsilon)(1 - \eta - e^{-\Omega(\eta^2)}) \max_{\mathbf{x} \in \mathcal{P}} F(\mathbf{x}) \geq (1 - \varepsilon - \eta - e^{-\Omega(\eta^2)}) \max_{\mathbf{x}_{OPT} \in \mathcal{C}} f(\mathbf{x}_{OPT}).$$

Since pipage rounding only converts $\hat{\mathbf{x}}$ into $\mathbf{x}'$ without requiring any information from $F$, it incurs no additional query complexity. Therefore, the overall query complexity remains

$$O\left(n^{1/2}\varepsilon^{-1/2}(T_1 + T_2)\right).$$

Pipage rounding is effective for both cardinality and partition matroid constraints. At the end of the rounding process, all coordinates are guaranteed to be integral. However, this is not the case for the knapsack constraint: in general, the process may leave one fractional coordinate, which requires a separate treatment.

During the rounding phase, let $\hat{\mathbf{x}}_0 \to \hat{\mathbf{x}}_1 \to \cdots \to \hat{\mathbf{x}}_k$ denote the sequence of intermediate fractional solutions produced by pipage rounding, where $\hat{\mathbf{x}}_0 = \hat{\mathbf{x}}$. At each step $t = 0, \ldots, k-1$, the convexity of the multilinear extension $F$ ensures that

$$\mathbb{E}[F_m(\hat{\mathbf{x}}_{t+1})] = \frac{-\theta_{\min}}{-\theta_{\min} + \theta_{\max}} F_m(\hat{\mathbf{x}}_t + \theta_{\max}\mathbf{v}) + \frac{\theta_{\max}}{-\theta_{\min} + \theta_{\max}} F_m(\hat{\mathbf{x}}_t + \theta_{\min}\mathbf{v}) \geq F_m(\hat{\mathbf{x}}_t).$$

By induction, we obtain

$$\mathbb{E}[F_m(\mathbf{x}_k)] \geq \mathbb{E}[F_m(\hat{\mathbf{x}})] = F_m(\hat{\mathbf{x}}).$$

According to Lemma 1

$$\mathbb{E}[F_m(\mathbf{x}_k)] \geq F_m(\hat{\mathbf{x}}) \geq \max_{\eta \in [0,1]} (1 - \eta) \left[1 - |V| \exp\left(-\eta^2 \triangle(\hat{\mathbf{x}})\right)\right] F(\hat{\mathbf{x}}).$$

where

$$\triangle(\hat{\mathbf{x}}) = \min_{i \in V} \frac{m_i \cdot \hat{\mathbf{x}}}{\max m_i} \geq \min_{i \in V} \frac{\min m_i \cdot B}{\max m_i \max c_i}.$$

$m_i \cdot \hat{\mathbf{x}} \geq \frac{\min m_i \cdot B}{\max c_i}$ holds because $\hat{\mathbf{x}}$ lies on the face of $\mathcal{P}$.

Thus, we have the following inequality:

$$\mathbb{E}[F_m(\mathbf{x}_k)] \geq \max_{\eta \in [0,1]} (1 - \eta) \left[1 - |V| \exp\left(-\eta^2 \min_{i \in V} \frac{\min m_i \cdot B}{\max m_i \max c_i}\right)\right] F(\hat{\mathbf{x}}).$$

Next, let $\mathbf{x}_{k+1}$ denote the integral vector obtained from $\mathbf{x}_k$ by keeping all entries equal to 1 unchanged, and rounding all fractional entries down to 0. Let $e = \arg\max_{e \in \mathcal{N}} f(e)$. Then we have

$$F_m(\mathbf{x}_k) - F_m(\mathbf{x}_{k+1}) \leq f(e),$$

which implies

$$\max\{F_m(\mathbf{x}_{k+1}), f(e)\} \geq \tfrac{1}{2}\left[F_m(\mathbf{x}_{k+1}) + f(e)\right] \geq \tfrac{1}{2}F_m(\mathbf{x}_k).$$

Taking expectation on both sides gives

$$\mathbb{E}[\max\{F_m(\mathbf{x}_{k+1}), f(e)\}] \geq \tfrac{1}{2}\mathbb{E}[F_m(\mathbf{x}_k)],$$

$$\geq \tfrac{1}{2} \max_{\eta \in [0,1]} (1 - \eta) \left[1 - |V| \exp\left(-\eta^2 \min_{i \in V} \frac{\min m_i \cdot B}{\max m_i \max c_i}\right)\right] F(\hat{\mathbf{x}}).$$

$$\geq \tfrac{1}{2} \max_{\eta \in [0,1]} (1 - \varepsilon)(1 - \eta) \left[1 - |V| \exp\left(-\eta^2 \min_{i \in V} \frac{\min m_i \cdot B}{\max m_i \max c_i}\right)\right] \max_{\mathbf{x} \in \mathcal{P}} F(\mathbf{x}).$$

Since

$$\max_{\mathbf{x} \in \mathcal{P}} F(\mathbf{x}) \geq \max_{\mathbf{x}_{OPT} \in \mathcal{C}} F(\mathbf{x}_{OPT}),$$

the inequality also holds under the knapsack constraint.

Since randomized rounding only converts $\hat{\mathbf{x}}$ into $\mathbf{x}'$ without requiring any information from $F$, it incurs no additional query complexity. However, the Best Singleton Selection step requires time complexity $O(n \cdot T_1)$. Therefore, the overall query complexity is

$$O\left(n \cdot T_1 + n^{1/2}\varepsilon^{-1/2}T_2\right).$$

APPENDIX E: CHOICE OF $\eta$

In Theorem 2, we establish that the approximation guarantee for the SCM maximization problem under cardinality and partition matroid constraints is $\max_{\eta \in [0,1]} (1 - \varepsilon)(1 - \eta) \left[ 1 - |V| \exp\left( -\eta^2 \min_{i \in V} \frac{\min m_i \cdot k}{\max m_i} \right) \right]$, and under the knapsack constraint it becomes $\frac{1}{2} \max_{\eta \in [0,1]} (1 - \varepsilon)(1 - \eta) \left[ 1 - |V| \exp\left( -\eta^2 \min_{i \in V} \frac{\min m_i \cdot B}{\max m_i \max c_i} \right) \right]$. A natural question is: *what is the appropriate choice of $\eta$?* If $\eta$ is too close to 0, the term $1 - |V| \exp\left( -\eta^2 \min_{i \in V} \frac{\min m_i \cdot k}{\max m_i} \right)$ becomes extremely small; if $\eta$ is too close to 1, the factor $(1 - \eta)$ collapses, making the overall expression small as well.

Figure 4 illustrates this trade-off: as $\eta$ increases from 0 to 1, the approximation ratio first increases and then decreases. Hence, selecting an appropriate $\eta$ that maximizes the bound is essential.

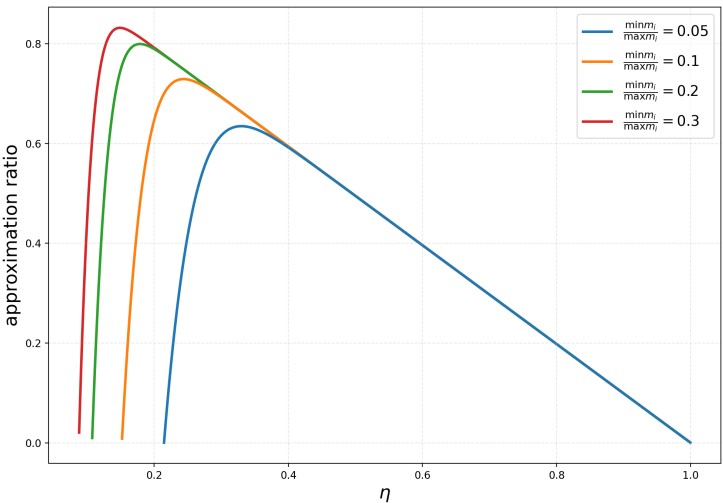

Figure 4: The figure illustrates the trends of the approximation ratios for cardinality constraints as $\eta$ varies, as stated in Theorem 2. Four curves corresponding to $\min m_i / \max m_i = 0.05, 0.1, 0.2, 0.3$ are plotted. The plot assumes $k = 1000$ and $|V| = 10$.

In Figure 5, we further examine the optimal choice of $\eta$ as the cardinality parameter $k$ and the knapsack capacity $B$ increase. We observe that, depending on the specific values of $k$ and $B$, the optimal $\eta$ may take almost any value within the interval $[0, 1]$. This highlights that selecting an appropriate $\eta$ is crucial for attaining a strong approximation guarantee.

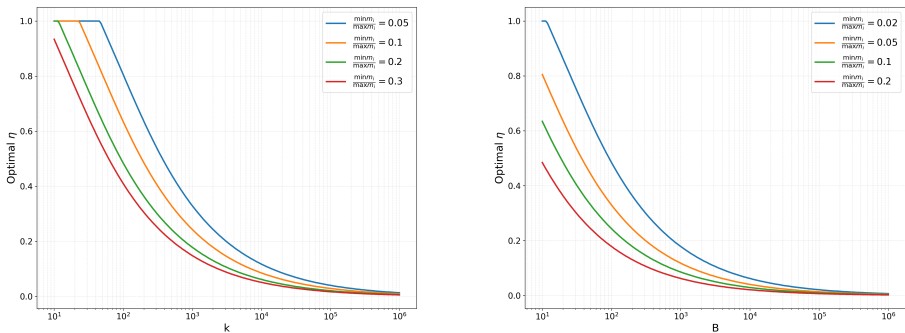

Figure 5: The left figure shows the optimal choice of $\eta$ for the cardinality constraint as $k$ varies, based on the bound in Theorem 2. Four curves are plotted for $\min m_i / \max m_i = 0.05, 0.1, 0.2, 0.3$, with $|V| = 10$. The right figure shows the optimal $\eta$ for the knapsack constraint as the capacity $B$ changes. Four curves are plotted for $\min m_i / \max m_i = 0.02, 0.05, 0.1, 0.2$, assuming $|V| = 10, \max c_i = 10$.

