# OpenReview forum: "Efficient Submodular Maximization for Sums of Concave over Modular Functions"
_ICLR.cc/2026/Conference — ICLR 2026 Poster_

### Official Review · Reviewer_suJC · 2025-10-30

**Soundness:** 1
**Presentation:** 1
**Contribution:** 2
**Rating:** 2
**Confidence:** 3

**Summary:**

In this paper, the authors study the problem of submodular maximization where the objective function can be expressed as the sum of concave composed with modular functions (SCMs), subject to various types of constraints. The proposed algorithms utilize the continuous relaxation, Accelerated Approximate Projected Gradient Ascent (AAPGA), and randomized rounding.  The method achieves an approximation ratio of $1-\epsilon-\eta-e^{-\Omega(\eta^2)}$ for both cardinality and partition matroid constraints and $1/2(1-\epsilon-\eta-e^{-\Omega(\eta^2)})$ for general matroid constraints.

**Strengths:**

1. The paper addresses an interesting problem of submodular maximization where the objective function is formulated as a composition of concave and linear functions. The motivation is clearly presented, and the inclusion of representative examples such as the coverage and facility location functions helps illustrate the relevance of the studied setting.
2. The authors tackle an important challenge in continuous optimization methods for submodular maximization. While such methods often yield stronger approximation guarantees compared to discrete approaches, their high query complexity and computational overhead have limited their practical use. Hence, the effort to design continuous algorithms that are both query-efficient and computationally efficient is timely and significant.
2. The integration of deep neural networks into the study of submodular optimization is an innovative and inspiring direction. This approach has the potential to be generalized to a broader class of submodular functions and could open new avenues for combining learning-based techniques with classical optimization theory.

**Weaknesses:**

The key concern is the poor clarity and organization of the paper. Several important details are missing from the main text, and some parameters are introduced without proper definition or explanation (see the Questions section for specifics). The core ideas underlying the proposed algorithms are not clearly presented, making it difficult to follow the methodological contributions. Furthermore, the paper does not clearly articulate how its technical contributions differ from or improve upon existing work.

**Questions:**

1. The parameter $\eta$ in Algorithm 1 is introduced without explanation, which makes it difficult to interpret the theoretical guarantees.
   - Is $\eta$ assumed to lie within the range \([0, 1]\)? If so, it seems that the approximation guarantee is valid only for certain values of \(\eta\), since when $\eta=0$ or $\eta=1$, the bound becomes trivial.
   - In addition, the scaling factor $\beta$ does not appear in the theoretical guarantee—are $\eta$ and $\beta$ intended to represent the same parameter?
   Clarification on this point is needed.
2. The definition of the set $P'_L(\mathbf{x})$ (Line 206) takes a point $\mathbf{x}$ as input; however, in the actual expression, only $y^{(t)}$ appears.
    It is unclear what $\mathbf{x}$ refers to in this context. Please clarify the definition and role of $\mathbf{x}$.
3. Can you clarify in detail why the proposed Accelerated Approximate Projected Gradient Ascent method is better than the existing PGA method?
4. The computational cost of \textbf{Algorithm 1} is not clearly analyzed. In particular, Line 5 in the pseudocode requires determining $i_t$, which involves evaluating Equation (4) for each possible $i_t$. How is the number of $i_t$ evaluations bounded?
5. The paper mentions the use of \textbf{deep neural networks} to accelerate the computation of the supergradient. However, if the objective function is simply a composition of a concave and a linear function, it is unclear why a neural network is necessary for this task.

---

> ### Author Response · Authors · 2025-11-17
> **Rebuttal by Authors**
>
> Weaknesses:
>
> The key concern is the poor clarity and organization of the paper. Several important details are missing from the main text, and some parameters are introduced without proper definition or explanation (see the Questions section for specifics). The core ideas underlying the proposed algorithms are not clearly presented, making it difficult to follow the methodological contributions. Furthermore, the paper does not clearly articulate how its technical contributions differ from or improve upon existing work.
>
> Due to the insufficient details provided in the paper (e.g., the computation of $i_t$), the reviewers may have found certain aspects of the paper unclear. Additionally, the paper involves several parameters that were not sufficiently explained. In the revised version, we have made improvements to address these issues and have attempted to answer the reviewer's questions in the relevant sections.
>
> On pages 2, lines 81 to 92, a brief description of two prior works was provided. Here, we offer a more detailed explanation. [1] considered the maximization problem of weighted coverage functions under cardinality and matroid constraints. Their approach introduced concave relaxation, followed by stochastic gradient ascent, and finally applied pipage rounding to achieve a $1 - 1/e$ approximation result. [2] further considered combining submodular functions with neural networks to solve the maximization problem under matroid constraints, which is an extension of the problem in [1]. The key difference in their method is the replacement of stochastic gradient ascent with projected gradient ascent (PGA), which yields an approximation ratio of $1 - \varepsilon - \eta - e^{(-\Omega(\eta^2))}$, with query complexity $O(n^2/\varepsilon^2 T_2)$, where $T_2$ is the computational cost of backpropagation. The issues with this method are as follows: first, PGA requires estimating the Lipschitz constant; second, it has poor convergence behavior; and third, it requires precise projections.
>
> To overcome these problem, we propose the Accelerated Approximate Projected Gradient Ascent (AAPGA) algorithm. First, in line 5 of Algorithm 1, we estimate the SCM's $L$ using a low-complexity algorithm, which allows us to dynamically adjust the step size. Second, in line 8 of Algorithm 1, we use the update information from the previous step to predict the direction of convergence for the next step, further accelerating the convergence rate. Third, in line 6 of Algorithm 1, we use an approximate projection to reduce the number of projections required. Therefore, our algorithm ultimately achieves a faster query complexity compared to [2] and performs better in terms of experimental results. Additionally, we design a randomized rounding technique, enabling the extension of the problem to the knapsack constraint.
>
> Other related work is listed in the table on line 54 of the revised manuscript.
>
> [1] Karimi, M., Lucic, M., Hassani, H., Krause, A. (2017). Stochastic submodular maximization: The case of coverage functions. Advances in Neural Information Processing Systems, 30.
>
> [2] Bai, W., Stafford Noble, W., Bilmes, J. A. (2018). Submodular maximization via gradient ascent: the case of deep submodular functions. Advances in neural information processing systems, 31.

---

> ### Author Response · Authors · 2025-11-17
> **Rebuttal by Authors**
>
> Question 1: The parameter $\eta$ in Algorithm 1 is introduced without explanation, which makes it difficult to interpret the theoretical guarantees.
> Is $\eta$ assumed to lie within the range ([0, 1])? If so, it seems that the approximation guarantee is valid only for certain values of $\eta$, since when $\eta=0$ or $\eta=1$, the bound becomes trivial.
> In addition, the scaling factor $\beta$ does not appear in the theoretical guarantee—are $\eta$ and $\beta$ intended to represent the same parameter?
> Clarification on this point is needed.
>
> (1) Origin and role of $\eta$.
> As discussed in Line 307 of the revised manuscript, the parameter $\eta$ was originally
> introduced in [2](Bai,2018) to quantify the gap between the multi-linear extension and the
> concave extension. $\eta$ domain is $[0,1]$, and it is a purely theoretical quantity that does not appear in the algorithm itself. Therefore, the purpose of $\eta$ is to
> characterize the strength of the theoretical approximation guarantee, and the analysis only
> requires selecting the value of $\eta$ that maximizes this guarantee.
>
> (2) how to optimize $\eta$.
> To clarify how $\eta$ should be chosen, we added two figures in Appendix E and an additional figure in the main text. As shown in Figure 4, the approximation ratio is unimodal in $\eta$: it increases for small $\eta$, reaches a peak, and then decreases. Thus, the optimal $\eta$ always lies in the interior of $[0,1]$. Figure 5 further shows that the optimal value depends on the problem scale. When $k$ or $B$ is small, the best $\eta$ is closer to $1$, whereas for larger $k$ or $B$, the optimal $\eta$ moves toward $0$. This behavior is consistent with the theoretical bounds in Figure 1, where larger $k$ or $B$ yields stronger approximation ratios (e.g., exceeding $1 - 1/e$ for cardinality constraints).
>
> (3) Why $\beta$ does not appear in the guarantees.
> The parameter $\beta$ is fundamentally different from $\eta$. Its role is to estimate the
> Lipschitz constant of the concave extension $F$ in our AAPGA method. Consequently, $\beta$ affects only the \emph{computational
> efficiency}—not the approximation ratio. As detailed in Lines 297--305 of the revised
> manuscript, incorporating $\beta$ introduces only an additional computational factor
> $\log(\beta L(f)/L_0)/\log \beta$, which does not change the query complexity.
> Therefore, $\beta$ is deliberately omitted from the theoretical guarantees.

---

> ### Author Response · Authors · 2025-11-17
>
> Question 2: The definition of the set (Line 206) takes a point as input; however, in the actual expression $P_{L}^{\prime}(\mathbf{x})$, only appears. It is unclear what refers to in this context. Please clarify the definition and role of $\mathbf{x}$.
>
> This is a typographical error. The projection is indeed performed on $\mathbf{y}^{(t)}$, not $\mathbf{x}$. We have corrected $P_{L}^{\prime}(\mathbf{x})$ to $P_{L}^{\prime}(\mathbf{y}^{(t)})$. Additionally, we have made the corresponding correction in Appendix A, Line 652, to ensure consistency across the document.
>
> Question 3: Can you clarify in detail why the proposed Accelerated Approximate Projected Gradient Ascent method is better than the existing PGA method?
>
> In [2] (Bai,2018), the PGA-based method requires $O(n^2 / \varepsilon^2)$ iterations to
> maximize a bounded concave function. In contrast, our AAPGA algorithm requires only
> $O(n^{1/2}\varepsilon^{-1/2})$ iterations, yielding a substantial acceleration. The main
> reason for this improvement is the use of Nesterov's accelerated gradient method. As shown in
> Line 8 of Algorithm 1, our update is not a standard gradient step; instead, it forms a convex
> combination of the current iterate and the previous update to predict a better search
> direction. This momentum-based extrapolation significantly accelerates convergence.
>
> Another key advantage comes from Line 5 of Algorithm 1, where we estimate the Lipschitz
> constant of the concave extension $F$ using $\bar{L}$. This allows us to choose more
> appropriate step sizes during optimization, further improving efficiency compared to the
> vanilla PGA method.
>
> From an empirical perspective, Section 4.1 compares the convergence behaviors of AAPGA and PGA. The results clearly show that AAPGA converges faster and achieves better performance in practice.
>
>
> Question 4: The computational cost of \textbf{Algorithm 1} is not clearly analyzed. In particular, Line 5 in the pseudocode requires determining $i_t$, which involves evaluating Equation (4) for each possible. How is the number of $i_t$ evaluations bounded?
>
> We understand the reviewer's concern, so we have provided a more detailed explanation of the query complexity calculation in Lines 297-305. There is no need to worry about the computational cost of $i_t$. Intuitively, the number of times that $\bar{L}$ needs to be increased is only a constant, and hence its computational cost is negligible in the overall complexity. For this reason, the contribution of $i_t$ does not affect the final query complexity.The detailed proof is as follows.
>
> The algorithm in Algorithm 1 computes the supergradient in Step 4 $T$ times, with each backpropagation step for the supergradient having a time complexity of $T_2$, resulting in a query complexity of $O(T \cdot T_2)$. In Step 5 of the algorithm in Algorithm 1, each iteration requires verifying whether $i_t = 0$ holds. The number of evaluations of $F\left(p_{\bar{L}}^\prime(\mathbf{y}^{(t)})\right)$ and $Q_{\bar{L}}\left(p_{\bar{L}}^\prime(\mathbf{y}^{(t)}),\, \mathbf{y}^{(t)}\right)$ is constant, requiring a query complexity of $O(T \cdot T_1)$. Since $\bar{L} \geq L(f)$ holds, we can deduce that $F\left(p_{\bar{L}}^\prime(\mathbf{y}^{(t)})\right) \geq Q_{\bar{L}}\left(p_{\bar{L}}^\prime(\mathbf{y}^{(t)}),\, \mathbf{y}^{(t)}\right)$ is satisfied, and thus $\bar{L}$ will not exceed $\beta L(f)$. As $\bar{L}$ grows exponentially, the number of iterations that do not satisfy $F\left(p_{\bar{L}}^\prime(\mathbf{y}^{(t)})\right) \geq Q_{\bar{L}}\left(p_{\bar{L}}^\prime(\mathbf{y}^{(t)}),\, \mathbf{y}^{(t)}\right)$ is given by $\log \left(\frac{\beta L(f)}{L_0}\right) / \log \beta$, which is of constant order. Therefore, the overall complexity is $T \cdot T_1 + T \cdot T_2 = O\left(n^{1/2}\varepsilon^{-1/2} (T_1 + T_2)\right)$.

---

> ### Author Response · Authors · 2025-11-17
> **Rebuttal by Authors**
>
> Question 5: The paper mentions the use of \textbf{deep neural networks} to accelerate the computation of the supergradient. However, if the objective function is simply a composition of a concave and a linear function, it is unclear why a neural network is necessary for this task.
>
> The use of deep neural networks to accelerate the computation of supergradients offers several advantages. First, due to the non-differentiability of concave functions, it is reasonable to use neural networks to compute the supergradient. If computed on a CPU, this would involve complex branching logic, significantly increasing the computational complexity. Furthermore, as mentioned in the Strengths section, combining learning-based techniques with classical optimization theory is a natural fit, as many optimization problems are represented using neural networks. Converting these functions to others and manually computing derivatives would be overly cumbersome.
>
> Additionally, using deep neural networks to solve for the supergradient is clearly faster than alternative methods. This can be empirically verified with an experiment. Consider the function $ f(x) = \sum_{j \in \mathcal{N}} \left( 1 - \exp\left( \sum_{i \in \mathcal{N}} x_i \log(1 - p_{ij}) \right) \right) $ as our submodular function. When computing the gradient of 32 vectors of dimension 100,000, the manual derivative $ \frac{\partial f}{\partial x_j} = - \exp\left( \sum_{i \in \mathcal{N}} x_i \log(1 - p_{ij}) \right) \cdot \log(1 - p_{ij}) $ takes 2.671 seconds. In contrast, using a neural network with GPU-based backpropagation takes only 0.096 seconds, significantly speeding up the computation. Thus, the use of deep neural networks in this context is not only reasonable but also highly efficient.
>
> The combination of submodular functions and neural networks holds great potential. For example, there have already been papers that first learn submodular functions and then use submodular maximization to solve the problem [3](Jaisankar, 2024). This direction is a promising area that remains to be further explored.
>
> [3] Jaisankar, V., Bandyopadhyay, S., Vyas, K., Chaitanya, V., Somasundaram, S. (2024). Postdoc: Generating poster from a long multimodal document using deep submodular optimization. arXiv preprint arXiv:2405.20213.

---

### Official Review · Reviewer_nxbx · 2025-10-31

**Soundness:** 2
**Presentation:** 3
**Contribution:** 3
**Rating:** 6
**Confidence:** 3

**Summary:**

This paper investigates the maximization problem of the sum of concave composed with modular functions (SCMs), an important subclass of submodular functions, under cardinality, knapsack, and partition matroid constraints respectively. Paper proposes an optimization framework that integrates three key components: continuous relaxation, accelerated approximate projected gradient ascent, and randomized rounding. The experimental evaluation assesses the effectiveness and acceleration advantages of our method from three perspectives: convergence speed, small-scale submodular maximization, and large-scale submodular maximization.

**Strengths:**

The paper introduces a novel optimization framework that employs the concave extension during the optimization phase to enhance computational efficiency, while leveraging the multilinear extension in the rounding phase to ensure theoretical approximation guarantees. The relationship between these two extensions is bridged by a key lemma, thereby balancing computational performance with theoretical assurance. Furthermore, to address the non-differentiability of certain activation functions in SCMs, the paper presents a supergradient calculation formula based on the right-hand derivative and integrates it with the backpropagation mechanism of neural networks, effectively resolving gradient computation in non-differentiable cases.

**Weaknesses:**

1. It is unclear whether the results for the randomized algorithms in Section 4.2 represent averages from multiple runs or the outcome of a single run.

2. In the experimental section, several key parameters that affect the performance of the AAPGA, such as the approximation projection error $\delta$ and the step size $L_0$, are not provided.

3. Some related works on submodular maximization under knapsack constraints may require further discussion to contextualize the current paper, such as:

   * *Fast adaptive non-monotone submodular maximization subject to a knapsack constraint*
   * *Submodular maximization subject to a knapsack constraint: Combinatorial algorithms with near-optimal adaptive complexity*
   * *Streaming algorithms for constrained submodular maximization*
   * *Linear-Time Algorithms for Representative Subset Selection From Data Streams*

**Questions:**

The paper discusses how to handle the non-differentiability of the activation function in SCM, but Theorem 1 assumes that $F$ is continuously differentiable. There seems to be a discrepancy between these two statements. Could you please clarify the rationale behind this assumption?

---

> ### Author Response · Authors · 2025-11-17
> **Rebuttal by Authors**
>
> Weaknesses 1: It is unclear whether the results for the randomized algorithms in Section 4.2 represent averages from multiple runs or the outcome of a single run.
>
> The code in Section 4.2 computes the average results and average time over 50 runs, as supplemented in the revised manuscript (Line 548, Page 9). Since only FAST and LS+PGB exhibit relatively large variances, while the variances for the other methods are almost zero, the variances are not listed in the table due to space limitations. The randomness has little impact on the final results.
>
> Weaknesses 2: In the experimental section, several key parameters that affect the performance of the AAPGA, such as the approximation projection error $\delta$ and the step size $L_0$, are not provided.
>
> $L_0$ is the initial Lipschitz constant, set to $L_0 = 1$, $\beta = 2$ here. Already added in (Line 478, Page 9). Experiments show that since the Lipschitz constant in AAPGA is a variable constant, setting $L_0 = 0.1, 1, 10$ has little effect on the results. However, for PGA, it is a fixed coefficient, so the parameter has a significant impact on the results. If $L$ is too small, the step size becomes too large, leading to oscillations in the results; if $L$ is too large, the step size becomes too small, resulting in slow convergence. Here, $\delta$ is as shown in Algorithm 3, and we need to ensure that $hi - lo < \delta / \sqrt{n}$. The pseudocode represents the theoretical case. In the actual algorithm, we terminate after 40 iterations. Therefore, if the knapsack coefficient $c_i$ is less than 1 and the dimension is 10000, $\delta$ is approximately $2^{-32}$. Due to the complexity of the algorithm details, the specific value of $\delta$ is not listed in the experiments.
>
>
> Weaknesses 3: Some related works on submodular maximization under knapsack constraints may require further discussion to contextualize the current paper, such as:
>
> Fast adaptive non-monotone submodular maximization subject to a knapsack constraint
>
> Submodular maximization subject to a knapsack constraint: Combinatorial algorithms with near-optimal adaptive complexity
>
> Streaming algorithms for constrained submodular maximization
>
> Linear-Time Algorithms for Representative Subset Selection From Data Streams
>
> We would like to express our gratitude to the reviewer for the references you provided. The first two papers primarily focus on the problem of maximizing non-monotone submodular functions under knapsack constraints, while the third paper mainly studies submodular maximization under \( k \)-system and \( d \)-knapsack constraints. In the fourth paper, the streaming algorithm for maximizing monotone submodular functions under knapsack constraints achieves a $1/2-\varepsilon$-approximation with a time complexity of \( O(n/\varepsilon \log 1/\varepsilon) \). This result is impressive and was overlooked in our previous literature review. It has now been included in the table on line 69 of page 2 and represents a very recent, nearly linear-time improvement. In addition, we also cite [1] to further enrich and complement our study.
>
> [1] Khuller, S., Moss, A., Naor, J. S. (1999). The budgeted maximum coverage problem. Information processing letters, 70(1), 39-45.
>
>
> Questions:
> The paper discusses how to handle the non-differentiability of the activation function in SCM, but Theorem 1 assumes that $F$ is continuously differentiable. There seems to be a discrepancy between these two statements. Could you please clarify the rationale behind this assumption?
>
> Thank you for pointing out the difference between the non-differentiability of the activation function and the smoothness assumption in Theorem 1. The inequality used in our analysis (see Line 662 in the revised manuscript) is guaranteed to hold only when $F$ has a Lipschitz-continuous gradient. For this reason, we assumed that $F$ is continuously differentiable with a Lipschitz gradient, which simplifies the analysis and allows us to derive clean theoretical guarantees.
>
> In practice, however, the SCMs we consider may involve activation functions that are not differentiable at certain points, so that the left and right derivatives of $F$ do not coincide and only supergradients exist. Nevertheless, as shown in the coverage function experiments in Sections 4.1 and 4.2, our method performs very well empirically even when $F$ is not differentiable everywhere.

---

> > ### Comment · Reviewer_nxbx · 2025-11-24
> >
> > Thank you for your rebuttal. After carefully reviewing your response and considering the feedback from other reviewers, I have decided to maintain my original score.

---

> ### Author Response · Authors · 2025-11-28
> **Rebuttal by Authors**
>
> We thank the reviewer for their insightful comments. This paper presents a highly promising approach to submodular maximization. Continuous methods are widely used for submodular optimization, but they have long been hindered by the high sampling complexity of the multilinear extension. As a result, most approximation guarantees remain purely theoretical and are difficult to reproduce in practical large-scale settings.
>
> By combining continuous relaxation, Accelerated Approximate Projected Gradient Ascent (AAPGA), and randomized rounding, our work reduces the complexity of cardinality-constrained SCM maximization to
> $O\left(n^{1/2}\varepsilon^{-1/2}(T_1 + T_2)\right),$
> achieving a sublinear query complexity. The proposed method leverages the structural properties of concave extensions and multilinear extensions to obtain further acceleration. In addition, thanks to GPU parallelism, both $T_1$ and $T_2$ can be computed significantly faster in practice. Our approach not only provides lower query complexity in theory but also demonstrates superior empirical performance.
>
> Since the method effectively bridges submodular optimization and neural network techniques while offering substantial improvements in computation speed, it has strong potential for future research. For example, one may train SCM functions using deep learning and then maximize them with our method. Submodular reinforcement learning is another promising direction in which our techniques can play a crucial role. In this sense, our work establishes a valuable connection between submodular optimization and artificial intelligence.
>
> If you find our clarifications and revisions satisfactory, we would be very grateful if you would consider revisiting your evaluation.

---

### Official Review · Reviewer_ehgF · 2025-10-31

**Soundness:** 3
**Presentation:** 3
**Contribution:** 3
**Rating:** 4
**Confidence:** 4

**Summary:**

This paper proposes an efficient algorithm, Accelerated Approximate Projected Gradient Ascent (AAPGA), for maximizing Sums of Concave composed with Modular functions (SCMs). The method applies a Nesterov-accelerated projected gradient ascent optimization directly on the function's concave extension. Theoretically, the algorithm achieves a $(1-\epsilon-\eta-e^{-\Omega(\eta^{2})})$ approximation guarantee with a "square-root level" query complexity of $O(n^{1/2}\epsilon^{-1/2}(T_{1}+T_{2}))$, where $T_1$ is the cost of backpropagation and $T_2$ is the cost of function evaluation. Empirical evaluations confirm that AAPGA converges faster and achieves superior results compared to the standard Projected Gradient Ascent (PGA) method.

**Strengths:**

1.	The paper considers a subclass of submodular functions and provides the algorithm with sublinear query complexity.
2.	The paper considers several important constraints in the submodular maximization area.
3.	Experimental results are provided to show the effectiveness of their algorithms.

**Weaknesses:**

1.	The meaning of the parameter $ \eta $ is barely explained in the abstract or the main text, which can be confusing to readers.
2.	The meanings of the parameter $T_1$ and $T_2$ are not consistent in the paper. It is confusing.
3.	The knapsack results seem weak: the rounding procedure sacrifices a $1/2$ approximation ratio by enumerating the largest item in an outer loop.

**Questions:**

1.	Can the costs of $T_1$ and $T_2$ be made explicit as closed-form expressions? If we consider the traditional value oracle model, what is $T_1$ and $T_2$？
2.	The meanings of the parameter $T_1$ and $T_2$ are not consistent in the paper. For example, let us consider the time of evaluation of the concave extension. In the abstract, it is $T_2$, but in the corollary 1, it is $T_1$. The parameters in table 1 and in the description of table 1 (in page 2) are also not consistent.
3.	What is the parameter $ \eta $? It seems this parameter appears in Lemma 1 which shows that $\eta$ is related to the SCM function itself. Thus if we cannot choose arbitrary small $\eta$, the approximation ratio of the proposed algorithm may be very bad.

---

> ### Author Response · Authors · 2025-11-17
> **Rebuttal by Authors**
>
> Weaknesses 1: The meaning of the parameter $\eta$ is barely explained in the abstract or the main text, which can be confusing to readers.
>
> Quesetion 3: What is the parameter $\eta$? It seems this parameter appears in Lemma 1 which shows that $\eta$ is related to the SCM function itself. Thus if we cannot choose arbitrary small $\eta$, the approximation ratio of the proposed algorithm may be very bad.
>
> As discussed in Line 307 of the revised manuscript, the parameter $\eta$ was originally
> introduced in [2](Bai,2018) to quantify the gap between the multi-linear extension and the
> concave extension. $\eta$ domain is $[0,1]$, and it is a purely theoretical quantity
> that does not appear in the algorithm itself. Therefore, the purpose of $\eta$ is to
> characterize the strength of the theoretical approximation guarantee, and the analysis only
> requires selecting the value of $\eta$ that maximizes this guarantee.
>
> As the reviewer correctly pointed out, $\eta$ cannot be arbitrarily small, since doing so would yield a very weak approximation guarantee. To clarify this point, we added two figures in Appendix E and an additional figure in the main text. As shown in Figure 4, the approximation ratio is unimodal in $\eta$: it increases for small $\eta$, reaches a peak, and then decreases. Thus, the optimal $\eta$ always lies in the interior of $[0,1]$. Figure 5 further shows that the optimal value depends on the problem scale. When $k$ or $B$ is small, the best $\eta$ is closer to $1$, whereas for larger $k$ or $B$, the optimal $\eta$ moves toward $0$. This behavior is consistent with the theoretical bounds in Figure 1, where larger $k$ or $B$ yields stronger approximation ratios (e.g., exceeding $1 - 1/e$ for cardinality constraints).
>
> Weaknesses 2: The meanings of the parameter $T_1$ and $T_2$ are not consistent in the paper. It is confusing.
>
> Questions 2: The meanings of the parameter $T_1$ and $T_2$ are not consistent in the paper. For example, let us consider the time of evaluation of the concave extension. In the abstract, it is $T_2$, but in the corollary 1, it is
> $T_1$. The parameters in table 1 and in the description of table 1 (in page 2) are also not consistent.
>
> Indeed, $ T_1 $ and $ T_2 $ were mixed up in the paper. In the paper, $ T_1 $ denotes the computational cost of evaluating the concave extension, and $ T_2 $ denotes the computational cost of backpropagation. Specifically, $ T_1 $ refers to the forward propagation, and $ T_2 $ refers to the backward propagation. In the introduction, $ O((n^2/\varepsilon^2)T_1) $ should be corrected to $ O((n^2/\varepsilon^2)T_2)$. We have now clarified all sections of the paper accordingly.
>
> Weaknesses 3: The knapsack results seem weak: the rounding procedure sacrifices a $1/2$ approximation ratio by enumerating the largest item in an outer loop.
>
> The main reason for the $1/2$ approximation ratio sacrifice is due to the rounding process, where a fractional term is left in the final step, which must be discarded, leading to a $1/2$ sacrifice. The scenarios where the knapsack-constrained submodular maximization algorithm can be applied are limited. In the numerical experiments of the paper, both the Stream algorithm and the Lazier-greedy algorithm have approximation ratios that do not exceed $1/2$. Algorithms with approximation ratios greater than $1/2$ typically face challenges in terms of query complexity, making them difficult to apply. Therefore, this algorithm represents a tradeoff that both satisfies theoretical guarantees and ensures time efficiency.
>
> Question 1: Can the costs of $T_1$ and $T_2$ be made explicit as closed-form expressions? If we consider the traditional value oracle model, what is $T_1$ and $T_2$？
>
> The computation on the GPU differs from the traditional value oracle model. For the sake of simplicity, we still adopt the concept of query complexity here. In the traditional value oracle model, costs such as $T_1 $ are often neglected, as the oracle is typically treated as a "black box," where the computational process is considered a "zero-cost" operation. As a result, we do not focus on the relationship between the oracle's computation process and the size of the ground set. Therefore, $ T_1 $ and $ T_2 $ do not appear in the traditional query complexity expression. If this causes confusion for the reviewer, we can treat $T_1 $ and $ T_2 $ as constants, i.e., $O(1) $. In this case, the query complexity for the cardinality constraint and partition matroid is $O(n^{1/2} \varepsilon^{-1/2})$, which is a sublinear algorithm; while for the knapsack constraint, the query complexity is $ O(n) $, which is a linear algorithm. In practice, computing the concave extension and backpropagation using neural networks on the GPU is much faster than computing the oracle on the CPU. Therefore, $ T_1$ and $ T_2 $ can be considered constants smaller than 1.

---

### Official Review · Reviewer_4AFT · 2025-11-01

**Soundness:** 3
**Presentation:** 3
**Contribution:** 4
**Rating:** 8
**Confidence:** 2

**Summary:**

This work studies submodular function optimization where the submodular function is a concave function over the sum of modular functions (SCM).  The authors establish a continuous algorithm for cardinality constraint,  whose approximation ratio is (1-\epsilon) and query complexity is $\sqrt(n) (T_1+T_2)$, where $T_1$ is the time needed for evaluating the concave function and $T_2$ is time needed for back propagation.  Authors present algorithms for knapsack case with an approximation ratio 1/2-\epsilon.

I would like to disclose that I have only superficial familiarity with continuous optimization techniques.

**Strengths:**

This work substantially improves the query complexity compared to previous works, especially for the cardinality constraint.  For example, for cardinality constraints, the best known query complexity is  $O(n/epsilon)$. Similarly, for knapsack, the best known quesry comlexity isO(nklog^2 n)$.

The main technical contribution is the development of the Accelerated Approximate Projected Gradient Ascent algorithm, that converges faster than the known algorithms.

**Weaknesses:**

While the query complexity is smaller, when measured in-terms of $n$, it is not clear what is effect of T_1 and T_2. If they are large, then it is not clear if the proposed algorithm offers any advantage.

Typically, continuous optimization algorithms run slowly compared to the discrete versions. Given that it is not clear how much computational advantage the proposed algorithm offers compared to the LS+PGB algorithm that uses $O(n/epsilon)$ queries

**Questions:**

Please see the weakness
What is the effect of T_1 and T_2? How large could they be in the worst-case?
It is not clear to me that the rounding techniques proposed are any different from the known pipage rounding. Can you explain the differences?
Can you compare the performance with the algorithms listed in lines 59, 60 and 61?

---

> ### Author Response · Authors · 2025-11-17
> **Rebuttal by Authors**
>
> Weaknesses 1: While the query complexity is smaller, when measured in-terms of $n$, it is not clear what is effect of $T_1$ and $T_2$. If they are large, then it is not clear if the proposed algorithm offers any advantage.
>
> Question 1: What is the effect of $T_1$ and $T_2$? How large could they be in the worst-case?
>
> This is a great question. In the traditional value oracle model, the query complexity does not account for the time spent on computing the oracle, nor does it investigate the relationship between the oracle's computation time and $ n $. Here, computing the concave extension and backpropagation can also be considered as a kind of "black box," where the computational cost is ignored. Therefore, for the cardinality constraint and partition matroid, the query complexity is $ O(n^{1/2} \varepsilon^{-1/2}) $, which is a sublinear algorithm, and for the knapsack constraint, the query complexity is $ O(n) $, which is a linear algorithm. Thus, the algorithm itself is quite fast. In practice, the concave extension and backpropagation are much faster than computing the oracle, so we can treat $ T_1 $ and $ T_2 $ as constants smaller than 1.
>
>
> Weaknesses 2: Typically, continuous optimization algorithms run slowly compared to the discrete versions. Given that it is not clear how much computational advantage the proposed algorithm offers compared to the LS+PGB algorithm that uses $O(n/\varepsilon)$ queries.
>
> Indeed, continuous optimization algorithms are often slower than their discrete counterparts.
> For example, under matroid constraints, the time complexity of continuous greedy methods can
> reach $O(n^7)$, making them impractical for real-world applications. The main source of this
> high complexity is the need to evaluate the multilinear extension, which requires sampling and
> approximately $O(n^3)$ operations for a single function evaluation. In contrast, our approach employs the concave extension, which avoids the computational burden
> associated with the multilinear extension. Moreover, our concave extension can be efficiently
> implemented using neural networks on GPUs, making its evaluation significantly faster in
> practice.
>
> Algorithms such as LS+PGB are highly randomized. Although LS+PGB benefits from parallelism and
> runs faster in wall-clock time, its solution quality is substantially worse. In addition, our
> query complexity is lower than that of LS+PGB, allowing our method to run efficiently under large-scale conditions.
>
>
> Question 1:  It is not clear to me that the rounding techniques proposed are any different from the known pipage rounding.Can you explain the differences?
>
> Under the cardinality and partition matroid constraints, the rounding technique used is pipage rounding. However, under the knapsack constraint, there is no pipage rounding base, so we introduce randomized rounding in the paper. After performing rounding under the knapsack constraint, a fractional value remains, which must be rounded to 0 in order to obtain an integer solution. As a result, randomized rounding incurs a $1/2$ sacrifice in the approximation ratio. This rounding technique is efficient, but it comes at the cost of some performance loss.
>
>
> Question 2: Can you compare the performance with the algorithms listed in lines 59, 60 and 61?
>
> As you pointed out, the query complexity of Lazier-Greedy, Fast, and LS+PGB is relatively high. In Section 4.2, we compare these algorithms with ours on small-scale problems. According to our experimental results, the performance of the Fast and LS+PGB parallel algorithms is generally inferior to that of the serial algorithms. Among the serial algorithms, the Lazier-Greedy algorithm stands out, balancing both time and efficiency. The Greedy algorithm also performs well, achieving very good results, but at the cost of longer computation time. Our algorithm demonstrates a significant speed and efficiency advantage under the cardinality constraint, which suggests that our algorithm is better suited for solving the SCM maximization problem.

---

### Meta-Review · Area_Chair_ACjf · 2026-01-15

**Summary:**

This paper tackles scalable maximization of SCM submodular objectives under cardinality/knapsack/partition matroid constraints via a continuous relaxation optimized with an accelerated approximate projected gradient method (AAPGA) plus rounding, with an emphasis on practical speedups through GPU-parallel evaluation/backprop. Reviewer support centers on significantly improved (oracle-style) query complexity and faster convergence than standard PGA, with strong practical motivation for large-scale settings; concerns focus on clarity/notation, the real impact of the T1 and T2 cost terms, and whether the knapsack rounding/guarantees are compelling.

**Reviewer Concerns:**

The rebuttal addresses most issues: it clarifies T1 (concave extension forward eval) vs T2 (backprop), fixes prior inconsistencies/typos, explains the role of 𝛽 as a purely theoretical gap parameter and shows how to choose it (unimodal behavior), and justifies why AAPGA improves over PGA (Nesterov acceleration + Lipschitz estimation + approximate projection). Experimental-protocol concerns (randomized methods averaged over 50 runs; key hyperparameters) are answered, and the knapsack rounding loss is framed as inherent/standard tradeoff. Remaining weaknesses are mainly presentation/organization (some reviewers still found it hard to follow) and that treating T1 and T2 as “small constants” is somewhat assumption-heavy without more concrete accounting in representative deployments; knapsack guarantees remain weaker than the cardinality/partition story.

**Reviewer Scores:**

4AFT: likely stays at 8, ehgF could move to 5 from 4, nxbx stays 6, amnd suJC might have moved up a little.

---

### Decision · Program_Chairs · 2026-01-26

Accept (Poster)